# TAMING CURVATURE: ARCHITECTURE WARM-UP FOR STABLE TRANSFORMER TRAINING

**Sameera Ramasinghe** *         **Ajanthan Thalaiyasingam**         **Hadi Mohaghegh Dolatabadi**

**Chamin Hewa Koneputugodage**         **Gil Avraham**         **Violetta Shevchenko**         **Yan Zuo**

**Karol Pajak**                     **Alexander Long**

Pluralis Research

## ABSTRACT

Training billion-parameter Transformers is often brittle, with transient loss spikes and divergence that waste compute. Even though the recently developed Edge of Stability (EoS) theory provides a powerful tool to understand and control the stability of optimization methods via the (preconditioned) curvature, these curvature-controlling methods are not popular in large-scale Transformer training due to the complexity of curvature estimation. To this end, we first introduce a fast *online* estimator of the largest (preconditioned) Hessian eigenvalue (i.e., curvature) based on a *warm-started* variant for power iteration with Hessian–vector products. We show theoretically, and verify empirically, that the proposed method makes per-iteration curvature tracking feasible at billion-parameter scale while being more accurate. Using this tool, we find that training instabilities coincide with surges in preconditioned curvature and that curvature grows with depth. Motivated by these observations, we propose *architecture warm-up*: progressively growing network depth to carefully control the preconditioned Hessian and stabilize training. Experiments on large Transformers validate that our approach enables efficient curvature tracking and reduces instabilities compared to existing state-of-the-art stabilization techniques without slowing down convergence.

## 1 INTRODUCTION

Scaling up Transformers has driven remarkable progress across domains, from large language models that power conversational systems to diffusion-based models for image generation (Vaswani et al., 2017; Kaplan et al., 2020; Brown et al., 2020; Ouyang et al., 2022; Ho et al., 2020; Rombach et al., 2022). Yet, despite these gains, large models frequently exhibit training instabilities, i.e, large loss spikes and even divergence, especially at scale (Chowdhery et al., 2022; Dehghani et al., 2023; Zhang et al., 2022; Molybog et al., 2023; OLMo et al., 2024). As billion-parameter training becomes the norm, improving training stability is paramount: transient instabilities, i.e., loss spikes, gradient blow-ups, or full divergence, can consume vast compute budgets and wall-clock time. As models and datasets scale, stabilizing optimization reduces monetary and environmental costs while improving reproducibility and throughput, enabling dependable progress.

Stabilization at scale often relies on *empirical controls* for attention and optimization: soft-capping the logits (Gemma Team, 2024), $QK$-normalization or $QK$-clip to bound dot-product magnitudes of queries and keys (Henry et al., 2020; Dehghani et al., 2023; Team et al., 2025), and learning-rate or batch warmup to temper early steps (Gilmer et al., 2021; Dubey et al., 2024). In parallel, the Edge of Stability (EoS) literature shows that gradient methods gravitate toward regions where the product of step size and curvature approaches the stability boundary from classical quadratic optimization theory: for full-batch Gradient Descent (GD), training spends long phases with $\eta \lambda_{\max}(H) \approx 2$ (Cohen et al., 2021; Wang et al., 2022)—where $\lambda_{\max}(H)$ and $\eta$ denotes the largest eigenvalue

---
*Corresponding author: sameera@pluralis.ai

of the Hessian $H$ and the step size, respectively—while for preconditioned/adaptive methods the relevant quantity is the *preconditioned*[1] curvature (Cohen et al., 2022; Damian et al., 2023). Thus the stability threshold is inversely proportional to the step size and directly governed by the largest eigenvalue of the (preconditioned) Hessian. Although many of the previous works in stabilizing Transformers can be interpreted as attempts to keep $\eta\,\lambda_{\max}(H)$ below this boundary (Zhai et al., 2023; Wortsman et al., 2023b; Gilmer et al., 2021; Shazeer & Stern, 2018), *verifying* such claims has been difficult in practice, because estimating the curvature online for billion-parameter Transformers remains memory and compute-intensive.

To this end, we first introduce an efficient method to estimate the curvature *online* using *warm-started power-iteration* with Hessian-Vector Products (HVP) tailored for large models. Our key insight is that the top eigenvector of the (preconditioned) Hessian is *slow-moving* and warm-starting with the previous step's eigenvector significantly 1) reduces the iteration count and 2) improves accuracy. In particular, we require *less than five HVPs per step* ($\lesssim 5$), an order of magnitude lower than existing methods (Granziol, 2025), while seamlessly extending to the time-varying preconditioned matrix for adaptive methods. We provide theoretical bounds for the change in the eigenvector and the resultant iteration saving. **This makes online curvature tracking feasible for billion-parameter Transformers**. We then use this approach to confirm that loss spikes in large-scale Transformers correlate with spikes in preconditioned curvature and show that the latter increases with the network depth.

Combining these insights, we introduce an *architecture warm-up* strategy for stable training. The idea is to ensure the (preconditioned) curvature follows the trend of the stability threshold such that the stability criterion is satisfied throughout training. Precisely, we restrict the model to have small curvature during the initial learning rate warm-up phase, and gradually relax this restriction (i.e., increase the curvature) when we start decaying the learning rate, noting that the stability threshold is inversely proportional to the learning rate. To control the curvature, we adopt a holistic approach of controlling the number of (effective) Transformer layers (i.e., depth), rather than making fine-grained modifications to each layer. Specifically, we freeze some Transformer layers to identity at initialization, and gradually unfreeze these layers as per a predefined schedule, ensuring a smooth increase in curvature. This architecture warm-up approach can be readily integrated to existing training recipes and standard architectures as it does not require dynamic computation graph surgery, outperforms existing stabilization techniques, and expands the range of stable learning rates without any performance penalty. We provide extensive experiments demonstrating accurate curvature tracking and consistent stability gains across large transformer settings compared to existing methods.

## 2 PRELIMINARIES

Below, we briefly review the literature on Edge of Stability (EoS) (Cohen et al., 2021; 2022), and power iteration to compute the largest eigenvalue of the Hessian using Hessian-Vector Products (HVP) (Martens, 2010), upon which we build our work. We refer the interested reader to the respective papers for more details.

### 2.1 EDGE OF STABILITY

For a quadratic objective $\mathcal{L}(\theta) = \frac{1}{2}\theta^\top A\theta + b^\top\theta + c$, gradient descent with step size $\eta$ is stable only if $\eta < 2/\lambda_{\max}(A)$. Locally, neural network training admits the quadratic approximation:

$$\mathcal{L}(\theta + \Delta) \approx \mathcal{L}(\theta) + \nabla\mathcal{L}(\theta)^\top\Delta + \tfrac{1}{2}\Delta^\top H(\theta)\Delta\,, \tag{1}$$

so the Hessian $H(\theta)$ plays the role of $A$, and $\lambda_{\max}(H(\theta))$ determines the maximum stable step size: violating $\eta \leq 2/\lambda_{\max}(H)$ causes oscillation or divergence along the sharpest direction. Empirically, full-batch GD often operates near the *Edge of Stability* (EoS) where $\eta\,\lambda_{\max}(H) \approx 2$ (Cohen et al., 2021). Adaptive methods (e.g., Adam (Kingma & Ba, 2014)) show an analogous behavior with the time-varying *preconditioned* curvature $\lambda_{\max}(P_t^{-1/2}HP_t^{-1/2})$ (Cohen et al., 2022), where $P_t^{-1}$ denotes the update preconditioning. For Adam, the preconditioner takes the

---

[1]Preconditioned Hessian should be considered for optimizers that use preconditioned updates, and adaptive methods are shown to operate at optimizer dependent stability thresholds (Cohen et al., 2022).

form: $P_t = \text{diag}(\sqrt{v_t} + \varepsilon)$ where $v_{t+1} = \beta_2 v_t + (1 - \beta_2)g_t^2$. Note that the stability criterion is optimizer-dependent and for Adam with $\beta_1 = 0.9$, adaptive EoS is determined to be $\eta\,\lambda_{\max}(P_t^{-1/2}HP_t^{-1/2}) \approx 38$.

This enables a powerful tool to understand and control the stability of optimization methods by controlling the learning rate and the preconditioned curvature. However, this theory has only been verified on small-scale models ($\lesssim 25$M parameters), mainly due to the memory complexity of computing the Hessian, or the time complexity associated with estimating an iterative approximation. Below, we first discuss a well-established power-iteration method to compute the curvature without materializing the Hessian explicitly, and later introduce our approach that reduces its iteration complexity by an order of magnitude, making online curvature tracking feasible and more accurate at billion-parameter scale.

## 2.2 Computing the Largest Hessian Eigenvalue via HVP-based Power Iteration

Given $\theta \in \mathbb{R}^d$ and the loss $f : \mathbb{R}^d \to \mathbb{R}$, we can estimate the top eigenpair $\{\lambda_{\max}(H(\theta)), v_{\max}(H(\theta)\}$, where $H(\theta) := \nabla^2 f(\theta)$ without explicitly forming $H(\theta)$. For any vector $v$,

$$H(\theta)v = \nabla_\theta\big(g(\theta)^\top v\big) = \left.\frac{d}{d\epsilon}\, g(\theta + \epsilon v)\right|_{\epsilon=0}. \tag{2}$$

Thus, $H(\theta)v$, i.e., the Hessian-Vector-product (HVP), is a *directional derivative* of the gradient (which quantifies how much the curvature would change in the direction of $v$) and can be obtained without materializing full $H$ (Martens, 2010). This costs roughly two backprop passes and uses $O(1)$ extra memory beyond the retained graph. The efficient computation of HVP allows us to compute the $\lambda_{\max}(H(\theta))$ and $v_{\max}(H(\theta))$ with power iteration. Let $H(\theta)$ be symmetric. Suppose its eigenvalues satisfy: $\lambda_1 \geq \lambda_2 \geq \cdots \geq \lambda_d, \qquad \rho := \frac{\lambda_2}{\lambda_1} \in [0, 1)$. Starting from a unit vector $y^{(0)}$, define the normalized power iteration:

$$z^{(t+1)} \leftarrow Hy^{(t)}\,, \qquad y^{(t+1)} \leftarrow \frac{z^{(t+1)}}{\|z^{(t+1)}\|_2}\,, \qquad \hat{\lambda}^{(t+1)} \leftarrow \langle y^{(t+1)}, Hy^{(t+1)}\rangle\,. \tag{3}$$

With above iterations, $z^{(t)} \to v_{\max}$ and $\hat{\lambda}^{(t)} \to \lambda_{\max}$ as $t \to \infty$. The only primitive is $u \mapsto Hu$, i.e., an HVP. This in practice is expensive for large $H$, since power iteration requires multiple steps to converge from a random initialization, each requiring two backward passes. This is more prominent in high dimensions where the initial alignment with the leading eigenvector is $O(1/\sqrt{d})$ in expectation, where $d$ is the dimension of the parameter vector.

## 3 Methodology

In this section, we first elaborate on our key insight that the (preconditioned) curvature of the Hessian is *slow-moving* with theoretical and empirical justifications, enabling us to develop a *warm-started power iteration* variant to compute the curvature efficiently with less than five HVP steps. This allows us to verify that the loss spikes in large-scale transformers are also a result of spikes in curvature, verifying the EoS theory at scale. Based on this, we later develop our *architecture warm-up* strategy that restricts the curvature of the model at the early learning rate warm-up phase and increases the curvature as the learning rate starts to decay, ensuring the stability criterion of adaptive EoS is satisfied throughout training.

### 3.1 Online Curvature Tracking with Warm-Started Power Iteration

We show that under practical assumptions, the largest eigenvalue of the Hessian of neural networks evolves slowly. Specifically, under Lipchitz continuity of the Hessian and a nonvanishing spectral gap $\gamma$, we can precisely bound the change in the leading eigenvector between successive steps:

$$\sin \angle\big(v_{1,k+1}, v_{1,k}\big) \leq \frac{L_H}{\gamma}\|\theta_{k+1} - \theta_k\|\,, \tag{4}$$

Thus, when step sizes and gradients shrink during training, $v_{1,k}$ is an increasingly accurate initializer for $v_{1,k+1}$. This result is formally presented below.

**Theorem 1.** *Let $\{\theta_k\}_{k\geq 0}$ be a parameter sequence and $H_k := H(\theta_k)$. Assume that there exists $L_H < \infty$ with $\|H(\theta) - H(\theta')\| \leq L_H\|\theta - \theta'\|$ for all $\theta, \theta'$. Equivalently, $\|\nabla^3 f(\theta)\|_{\mathrm{op}} \leq L_H$ and Along the parameter path considered, $\gamma(\theta) := \lambda_1(H) - \lambda_2(H) \geq \gamma > 0$. Then, with $v_{1,k}$ a unit top eigenvector of $H_k$ and $\varepsilon_k := \angle(v_{1,k+1}, v_{1,k})$, we have*

$$\sin\varepsilon_k \;\leq\; \frac{\|H_{k+1} - H_k\|}{\gamma} \;\leq\; \frac{L_H}{\gamma}\,\|\theta_{k+1} - \theta_k\|\;. \tag{5}$$

*Further, with stochastic gradient descent with step size $\eta_k$ such that $\theta_{k+1} = \theta_k - \eta_k g_k$ with $\mathbb{E}[g_k \mid \theta_k] = \nabla f(\theta_k)$ and $\mathbb{E}\|g_k\|^2 \leq G^2$, we have*

$$\mathbb{E}\big[\sin\varepsilon_k \mid \theta_k\big] \leq (L_H/\gamma)\,\eta_k\,\mathbb{E}\|g_k\| \leq (L_H/\gamma)\,\eta_k G\;. \tag{6}$$

Leveraging the above result, we propose a simple yet novel modification: **warm-starting power iteration across training steps**. At iteration $k$, suppose we have obtained an estimate of the top eigenvector $v_{1,k}$ of the Hessian $H_k = \nabla^2 f(\theta_k)$. At the next training iteration, instead of reinitializing power iteration from a random vector, we initialize from $v_{1,k}$ and run power iteration on $H_{k+1}$. Below, we quantify the gain in iteration count due to warm-start.

**Theorem 2.** *Let $H_k = H(\theta_k)$ have eigenvalues $\lambda_{1,k} \geq \lambda_{2,k} \geq \cdots$ and unit eigenvectors $v_{i,k}$. Set $\rho_{k+1} := \lambda_{2,k+1}/\lambda_{1,k+1} \in [0,1)$. Define the successive misalignment $\varepsilon_k := \angle(v_{1,k+1}, v_{1,k})$. Run normalized power iteration on $H_{k+1}$:*

$$y^{(t+1)} = \frac{H_{k+1}y^{(t)}}{\|H_{k+1}y^{(t)}\|}\;, \qquad y^{(0)} = v_{1,k}\;, \tag{7}$$

*and let $\alpha_t := \angle(y^{(t)}, v_{1,k+1})$. Then:*

$$0 \;\leq\; \lambda_{1,k+1} - y^{(t)\top}H_{k+1}y^{(t)} \;\leq\; \big(\lambda_{1,k+1} - \lambda_{2,k+1}\big)\sin^2\alpha_t \;\leq\; (1 - \rho_{k+1})\,\lambda_{1,k+1}\,\rho_{k+1}^{2t}\tan^2\varepsilon_k. \tag{8}$$

*Also, let $t$ and $t_{rand}$ be the number of iterations needed for warm start and random initialization to achieve convergence, respectively. Then, with a high probability, we have,*

$$t_{rand} - t \;\approx\; \frac{\frac{1}{2}\log d - \log\big(\frac{L_H}{\gamma}\|\theta_{k+1} - \theta_k\|\big)}{\log(1/\rho_{k+1})}\;. \tag{9}$$

Hence warm-starting is strictly advantageous whenever $\frac{L_H}{\gamma}\|\theta_{k+1} - \theta_k\| \ll d^{1/2}$.

### 3.1.1 EXTENSION TO THE PRECONDITIONED HESSIAN

When the optimization algorithm incorporates momentum and adaptive learning-rate scaling (i.e, preconditioning), stability depends on the *preconditioned* curvature. Let us consider the Adam update: $\theta_{t+1} = \theta_t - \eta\,P_t^{-1}m_{t+1}$, where $m_t$ is the momentum, updated as an exponential moving average $m_{t+1} = \beta_1\,m_t + (1 - \beta_2)\,g_t$ with $g_t = \nabla\mathcal{L}(\theta_t)$ and the preconditioner is the square-root of the second moment of the gradients. The *effective* curvature, therefore, is the spectrum of

$$G_t \;:=\; P_t^{-1/2}\,H(\theta_t)\,P_t^{-1/2}\;, \tag{10}$$

not of $H(\theta_t)$ itself. Because the preconditioner $P_t$ changes slowly when $\beta_2 \approx 1$ and $H(\theta_t)$ is Lipschitz smooth, $G_t$ evolves smoothly along the training trajectory. As a result, our warm-starting extends verbatim to the preconditioned Hessian. That is, the previous warm-start analysis for $H(\theta)$ carries over by replacing $H$ with $G_t$. Define the top eigenvector $u_{1,t}$ of $G_t$ and the eigengap $\gamma_t^{\mathrm{eff}} = \lambda_1(G_t) - \lambda_2(G_t)$. From Theorem 1 we can directly obtain:

$$\sin\angle\big(u_{1,t+1}, u_{1,t}\big) \;\leq\; \frac{\|G_{t+1} - G_t\|_2}{\gamma_t^{\mathrm{eff}}}\;. \tag{11}$$

So under a Lipschitz Hessian ($\|H(\theta_{t+1}) - H(\theta_t)\| \leq L_H\|\theta_{t+1} - \theta_t\|$) and slowly varying $P_t$ (e.g., $\|P_{t+1}^{-1/2} - P_t^{-1/2}\| \leq L_P\|\theta_{t+1} - \theta_t\|$ with $L_P \propto (1 - \beta_2)$), the top eigendirection of $G_t$ is *slow-moving*. Consequently, power iteration on $G_{t+1}$ *warm-started* from $u_{1,t}$ enjoys the same benefits as in the non-preconditioned case. This allows us to compute HVP for $G_t$ without new primitives.

**Warm-started tracking of the preconditioned Hessian.** At step $t$, we form the diagonal pre-conditioner $P_t = \text{diag}(\sqrt{v_t} + \varepsilon)$ from the optimizer state. We estimate the top eigenpair of $G_t := P_t^{-1/2} H(\theta_t) P_t^{-1/2}$ by power iteration with a *warm start* from the previous step. Concretely: (i) initialize $y^{(0)}$ with the transported eigenvector $y^{(0)} = \text{normalize}\left(P_t^{1/2} P_{t-1}^{-1/2} u_{1,t-1}\right)$ (or simply $y^{(0)} = u_{1,t-1}$ if transport is omitted); (ii) for $\tau = 0, 1, \ldots$ perform one *preconditioned HVP* power step

$$u = P_t^{-1/2} y^{(\tau)}, \quad v = H(\theta_t)\, u \ (\text{HVP}), \quad z = P_t^{-1/2} v, \quad y^{(\tau+1)} = z/\|z\|_2 ; \quad (12)$$

(iii) compute the Rayleigh estimate $\hat{\lambda}_t = (y^{(\tau)})^\top G_t y^{(\tau)} = u^\top v$ and stop when the change in $\hat{\lambda}_t$ falls below a tolerance. The output $(\hat{\lambda}_t, u_{1,t} = y^{(\tau^\star)})$ is reused at step $t+1$. Each iteration costs a single HVP plus cheap elementwise scalings by $P_t^{\pm 1/2}$; with warm starts, $\tau^\star$ is typically $< 5$, enabling efficient, step-by-step tracking of the *effective* curvature that governs stability under momentum/Adam (e.g., see Fig. 3).

## 3.2 ARCHITECTURE WARM-UP

Now, we are ready to introduce our approach to control the curvature. Specifically, we first review that deeper networks have the potential to increase the curvature, and provide an approach to seamlessly increase the number of trainable layers without introducing any function or gradient discontinuities. Recall that, at the EoS, the learning rate and curvature are inversely proportional; therefore, we keep the network shallow in the early learning rate warm-up phase, and gradually increase (effective) depth when we start decaying the learning rate, ensuring the stability criterion is satisfied throughout training. The proposed method can be readily integrated into existing training recipes and assumes a standard transformer architecture without requiring hardware-level operations.

### 3.2.1 DEEPER NETWORKS INCREASE CURVATURE AND SHRINK STABILITY THRESHOLD

Let $g_\theta = \Phi_L \circ \cdots \circ \Phi_1$ be an $L$-block residual Transformer with $\Phi_\ell(x) = x + B_\ell(x)$. For the input Jacobian,

$$\|J_x g_\theta(x)\| \leq \prod_{\ell=1}^{L} \|I + \partial B_\ell(x_{\ell-1})\| \leq \exp\left(\sum_{\ell=1}^{L} \|\partial B_\ell(x_{\ell-1})\|\right). \quad (13)$$

For losses $\mathcal{L}(\theta) = \mathbb{E}_{(x,y)}[\ell(g_\theta(x), y)]$ with $\lambda_{\max}(\nabla_z^2 \ell) \leq L_\ell$, the Gauss–Newton bound yields

$$\lambda_{\max}(\nabla_\theta^2 \mathcal{L}(\theta)) \leq L_\ell \left(\sup_x \|J_\theta g_\theta(x)\|\right)^2, \quad (14)$$

and $J_\theta g_\theta$ inherits the multiplicative growth in Eq. 13 through backprop. Hence $\lambda_{\max}(H)$ (and, under a Positive Semi-Definite (PSD) diagonal preconditioner $P_t$, $\lambda_{\max}(G_t)$) increases with depth $L$, shrinking the first-order stability margin. See App. B for more formal results and discussions.

**Remarks.** Although we can derive an upperbound as in Eq. 14 it is not sufficient to conclude that curvature must strictly increase with depth. Empirically, the curvature might be lower than the bound due to architecture components such as residual connections, and normalization (Sagun et al., 2016; Li et al., 2018; Ghorbani et al., 2019; Yao et al., 2020), or due to the specific parameterization of the model (Dinh et al., 2017). Furthermore, since the *preconditioned* geometry is the relevant one for adaptive optimizers, efficient *online* tracking of (curvature) top eigenvalues, especially the preconditioned analogue $\lambda_{\max}(G_t)$, is more informative than pointwise analyses at initialization or at local optima (Yao et al., 2020; Sagun et al., 2016; Ghorbani et al., 2019). Our method enables such tracking up to multi-billion-parameter Transformers, where we observe (in Sec. 4.3) that (i) preconditioned curvature spikes predominantly during learning-rate warm-up and (ii) increases with depth.

### 3.2.2 PROGRESSIVE DEPTH VIA CONSTRAINING BLOCK WEIGHTS TO ZERO

Recall that at the stability threshold, the (preconditioned) curvature $\lambda_{\max}(G_t)$ scales inversely with the learning rate $\eta$: larger $\eta$, the *smaller* the admissible $\lambda_{\max}(G_t)$ must be to remain stable. Consequently, while $\eta$ ramps up during warmup (tightening the threshold), we keep the network's *effective depth* low to limit $\lambda_{\max}(G_t)$. As training proceeds—after the peak learning rate or during

learning-rate decay, when the stability threshold relaxes—we progressively enable additional depth, activating the full model only once the stability margin permits it. To this end, we wish to add depth keeping *controlling curvature* and avoiding function discontinuities. Note that the transformer block function can be expanded as follows: Let the input to the $l^{\text{th}}$ layer be $\mathbf{X}^l \in \mathbb{R}^{b \times n \times d}$, where $b$, $n$, and $d$ are the batch size, sequence length, and embedding dimension, respectively. Then the function admits a recursive structure as

$$\mathbf{X}^{l+1} = \mathbf{X}^l_{\text{hidden}} \mathbf{W}^l_{p_2} + \mathbf{X}^l_{\text{concat}} \mathbf{W}^l_{p_1} + \mathbf{X}^l \, , \tag{15}$$

where hidden and concat are the hidden layer of the feedforward network and the concatinated output of the attention heads, respectively. $\mathbf{W}^l_{p_1}$ and $\mathbf{W}^l_{p_2}$ are linear projection weights. A natural idea is to hold only the projection matrices at zero (i.e., $\mathbf{W}^l_{p_1} = \mathbf{W}^l_{p_2} = \mathbf{0}$) so the block computes the identity with $\mathbf{X}_{l+1} = \mathbf{X}_l$, and can later be "unlocked." However, this can still induce a *discontinuous jump in the network's Lipschitz constant* at unlock: even a small change in $\mathbf{W}^l_{p_1}$ or $\mathbf{W}^l_{p_2}$ immediately injects the pre-existing (randomly initialized) attention/FFN paths into the residual stream. Formally, the input–output Jacobian of the $l$-th block satisfies the first-order bound

$$\|J_{\mathbf{X}^l} \Phi_l - I\| \ \leq \ \|\mathbf{W}^l_{p_1}\| \, \|J_{\mathbf{X}^l} \mathbf{X}^l_{\text{concat}}\| \ + \ \|\mathbf{W}^l_{p_2}\| \, \|J_{\mathbf{X}^l} \mathbf{X}_{\text{hidden}}\|. \tag{16}$$

If all other weights retain their (nonzero) random initialization while the projections are zeroed, then $\|J_{\mathbf{X}^l} \mathbf{X}^l_{\text{concat}}\|$ and $\|J_{\mathbf{X}^l} \mathbf{X}_{\text{hidden}}\|$ can already be large at the moment of unlock; consequently, even tiny updates to $\mathbf{W}^l_{p_1}$ or $\mathbf{W}^l_{p_2}$ can *abruptly raise* $\|J_{\mathbf{X}^l} \Phi_l\|$, inflating $\sup_x \|J_\theta g_\theta(x)\|$ of the bound Eq. 14, and thus increasing the curvature. This sudden elevation can push the model over the stability threshold, often manifesting as loss spikes or instabilities.

To remedy this, and to guarantee continuity of both the *function* and its *first derivative* at unlock, we set all the weights (except RMSNorm weights)[2] and exclude these parameters from the optimizer while the block is locked. Under this constraint, $\mathbf{X}^l_{\text{concat}} = \mathbf{0}$ and $\mathbf{X}_{\text{hidden}} = \mathbf{0}$, so $\mathbf{X}^{l+1} = \mathbf{X}^l$ and $J_{\mathbf{X}^l} \Phi_l = I$, i.e., the block is an exact identity with *no increase* in the Jacobian. When the block is unlocked, all paths start from zero, and the Jacobian perturbation grows smoothly as the newly trainable weights move away from zero. This prevents the instantaneous jump in effective Jacobian and thus avoids the associated curvature spike. In practice, we keep all block weights at zero and frozen within the optimizer until a curvature criterion is met; then we start training them with zero initialization. This *architecture warm-up* keeps the product $\eta \, \lambda_{\max}(G_t)$ within the stability envelope while depth increases, yielding smoother loss and more reliable training.

**Does architecture warm up compromise representation capacity?** We provide intuitions from two perspectives; (i) *Spectral bias / F-principle:* deep nets fit *low*-frequency structure first, with higher frequencies learned later (Rahaman et al., 2019; Xu et al., 2019); a shallow stack suffices early, so temporarily limiting depth does not bottleneck what the model actually learns. (ii) *Function-space/NTK view:* early training operates in a near-linear, low-curvature regime where learning aligns with dominant, low-complexity components (Jacot et al., 2018); additional layers can be enabled later to increase expressivity without restricting the attainable solution class. Our convergence results, and prior work on progressive, function-preserving growth, corroborate that deferred depth does not harm final performance (Chen et al., 2016; Wei et al., 2016; Gong et al., 2019; Chen et al., 2020).

## 4 EXPERIMENTS

We evaluate decoder-only Transformers (Llama 3–style (Dubey et al., 2024)) on three large-scale corpora: Fineweb (Penedo et al., 2024), DCLM (Li et al., 2024) and Olmo-Mix (Allen Institute for AI, 2024). We reserve a held-out set for validation. Unless otherwise noted, we train with context length 1024, embedding dimension 2048, 32 heads, global batch size 1024, weight decay 0.01, AdamW optimizer with standard parameters, and a linear warmup over 2000 steps followed by linear decay. Tokenization is GPT-2 (vocabulary size 50,000). We use five power iterations with warm-start for online curvature tracking. Our experiments include models scaling up to 3-billion parameters. **Architecture warm-up schedule:** Unless noted otherwise, we keep the model at half depth until learning-rate warmup completes, then unlock the remaining layers in four groups,

---

[2]Excluding RMSNorm weights is critical since they show inferior convergence from zero initialization.

spaced by 500 training iterations, to reach full depth. Although we use this schedule, we observed that performance is not tightly coupled to this spacing (see App. E).

## 4.1 SLOWLY MOVING TOP EIGENVECTORS

Theorem 1 predicts that the leading Hessian eigendirection moves slowly along the optimization path. To verify this empirically (without any estimator bias), we compute the *exact* Hessian for a 4-layer Transformer and measure the principal angle between successive top eigenvectors across training steps. We choose a shallow network as computing the exact Hessian of a large model is computationally prohibitive. Fig. 1 shows that these angles are typically $< 0.1$ rad ($\approx 5.7°$) (except for a few spikes), confirming the "slow drift" property.

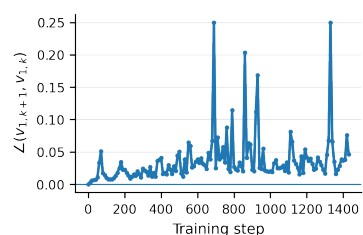

Figure 1: **Leading eigenvector is slow-moving.** For a 4-layer Transformer, we compute the *exact* top Hessian eigenvector at each train step and plot the principal angle to the next step. Consistent with Theorem 1, angles are typically $< 0.1$ radians, with only rare outliers.

## 4.2 EFFECTIVENESS OF WARM-START POWER ITERATION

We compare our warm-started estimator to a cold-start (random) power method on a 4-layer model, using the relative error $|\hat{\lambda} - \lambda_{\text{exact}}|$ as the metric. Figure 2 reports error versus iteration count. We ran each experiment five times and report the error bars. Warm start achieves high-accuracy estimates even with $\approx 5$ iterations, while the cold start error is higher with even $\geq 20$ iterations. Further, the variance of the cold-start error is constantly high across the number of iterations. As shown in Fig. 2 this is due to the occasional large errors at certain steps (sensitivity to the initializer) even with a high iteration count. This confirms that reusing the previous step's top direction substantially reduces the HVP budget and stabilizes power convergence.

## 4.3 DEPTH, EFFECTIVE CURVATURE, AND STABILITY

We examine the impact of depth on curvature and stability by training 8, 16, and 32-layer models on the FineWeb dataset with peak learning rate $8 \times 10^{-3}$. Figure 3 plots $\lambda_{\max}(H)$ and $\lambda_{\max}(G_t)$ over training. The 8-layer network maintains low, stable curvature and a smooth loss trajectory. As depth increases, both $\lambda_{\max}(H)$ and $\lambda_{\max}(G_t)$ exhibit higher levels and variability, and the training loss becomes prone to spikes and divergence. These results support the hypothesis that deeper stacks go beyond the stability margin with increased curvature. This observation motivates our *architecture warm-up*: start shallow (low curvature), when during the learning rate warm-up (where the stability threshold increasingly becomes lower) and progressively unlock additional layers when the stability margin is higher.

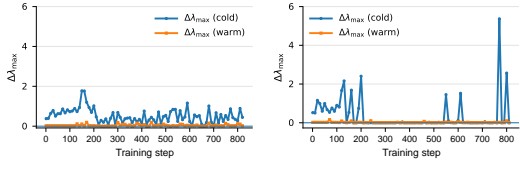

(a) 5 HVP iterations.  (b) 20 HVP iterations.

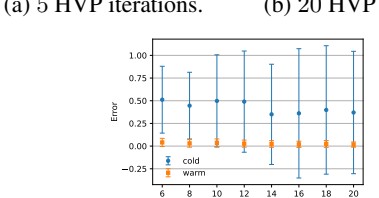

(c) Error across different # HVP iterations.

Figure 2: **Effectiveness of warm-start HVP.** (a,b) Online tracking error (vs. exact curvature) using 5 and 20 power-iteration steps, respectively, over the course of training. (c) Error vs. number of power-iteration steps with error bars over 5 random seeds (both model init and HVP probes). The warm-started estimator converges faster (even with 5 HVPs) and attains lower error and variance than cold-start baselines.

## 4.4 STABILITY OF ARCHITECTURE WARM-UP

As discussed in Sec. 3.1.1, under the stability threshold, $\lambda_{\max}(G_t)$ scales as $O(1/\lambda_{\max}(G_t))$ against $\eta$ in first-order methods. Thus, larger $\eta$ demands *smaller* effective curvature. We show that *architecture warm-up*, which suppresses

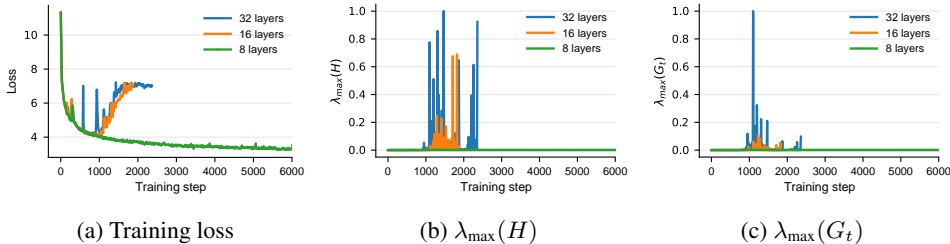

(a) Training loss       (b) $\lambda_{\max}(H)$       (c) $\lambda_{\max}(G_t)$

Figure 3: **Curvature vs. depth.** We train transformers of increasing depth/size—8 layers (640M), 16 layers (1B), and 32 layers (3B)—with a learning rate $8 \times 10^{-3}$ and track training loss, $\lambda_{\max}(H)$, and $\lambda_{\max}(G_t)$. Consistent with Sec. 3.2.1, curvature rises with depth: the 16 and 32-layer models exhibit pronounced spikes in $\lambda_{\max}(H)$ and $\lambda_{\max}(G_t)$ during learning-rate warmup, crossing the stability boundary and diverging. By contrast, the 8-layer model maintains lower curvature and converges stably ($\lambda_{\max}(H)$ and $\lambda_{\max}(G_t)$ are normalized).

curvature early and lets it grow in a controlled manner, substantially widens the range of stable learning rates: across peak-$\eta$ sweeps, models trained with architecture warm-up maintain bounded $\lambda_{\max}(G_t)$ and avoid loss spikes, whereas vanilla networks (no architecture warm-up) exhibit rapid curvature escalation and unstable convergence under the same settings (Fig. 4).

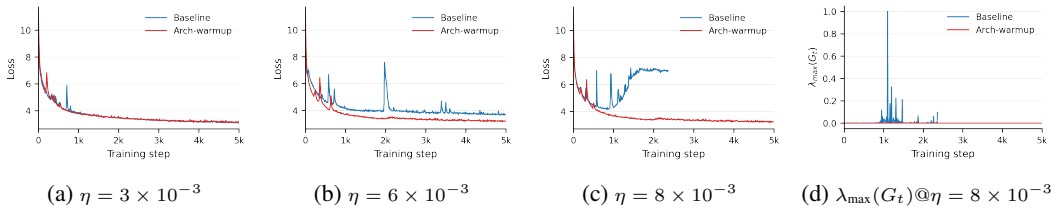

(a) $\eta = 3 \times 10^{-3}$    (b) $\eta = 6 \times 10^{-3}$    (c) $\eta = 8 \times 10^{-3}$    (d) $\lambda_{\max}(G_t)@\eta = 8 \times 10^{-3}$

Figure 4: **Convergence under varying stability thresholds.** As the stability boundary scales as $O(1/\eta)$ for fixed curvature, we sweep the peak learning rate $\eta$ (i.e., tightening/relaxing the threshold) and compare *architecture warm-up* to an unaltered baseline. As $\eta$ increases (smaller $1/\eta$ margin), the baseline becomes increasingly unstable, whereas architecture warm-up maintains stable convergence across the range. Models are trained on FineWeb.

## 4.5 COMPARISON AGAINST OTHER STABILIZATION METHODS

We compare architecture warmup to three state-of-the-art stabilization baselines, QK-Norm (Henry et al., 2020), QK-Clip (Team et al., 2025), and Softcap (Gemma Team, 2024), and an unmodified baseline (Fig. 5) with a peak LR of $8 \times 10^{-3}$. To this end, we use 16-layer, 1B parameter models. Note that we intentionally use a higher learning rate to observe the performance of methods under a smaller stability threshold. Architecture Warmup consistently reduces spike frequency and magnitude, avoids divergence, and exhibit faster convergence where other methods destabilize, yielding more reliable training across datasets. Interestingly, we found QK-CLIP to be quite unstable, often diverging to NaN values mid training. Table 1 shows validation perplexities. As QK-norm was performing best on FineWeb, we compare against it on a longer training run, up to Chinchilla (Hoffmann et al., 2022) compute optimal (see App. D)

Table 1: Perplexity ($\downarrow$) across three datasets and five methods. $*$ indicates the diverged runs.

| Dataset | Baseline | QK-Norm | QK-Clip | Softcap | Arch-Warmup |
|---------|----------|---------|---------|---------|-------------|
| FineWeb | $*$ | $49.88\pm0.003$ | $*$ | $51.41\pm0.002$ | $\mathbf{25.02 \pm 0.001}$ |
| DCLM | $165.62\ \pm0.04$ | $61.57\pm0.012$ | $*$ | $43.38\pm0.03$ | $\mathbf{22.64 \pm 0.002}$ |
| OLMo-Mix | $*$ | $*$ | $*$ | $*$ | $\mathbf{18.54 \pm 0.001}$ |

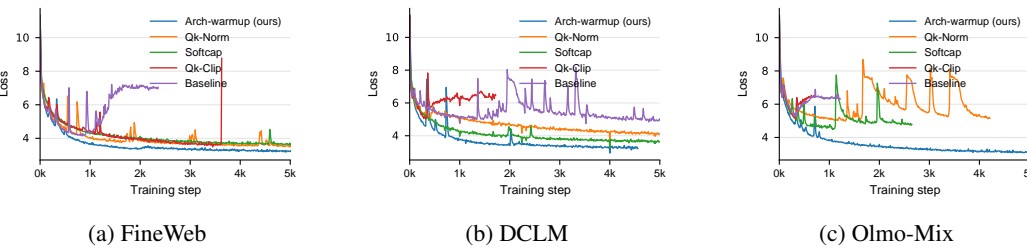

(a) FineWeb       (b) DCLM       (c) Olmo-Mix

Figure 5: **Comparison against existing stabilization techniques.** Across datasets, competing methods converge more slowly and exhibit frequent loss spikes, sometimes leading to divergence, whereas our method remains stable and consistently faster to train.

### 4.6 CAN LEARNING-RATE WARMUP BE REPLACED BY ARCHITECTURE WARM-UP?

Learning-rate warmup stabilizes early training by enlarging the stability margin ($\propto O(1/\eta)$) when curvature is high; as $\eta$ increases, this margin tightens. *Architecture warm-up* plays the complementary role by directly *controlling curvature*: keep depth (and thus $\lambda_{\max}(G_t)$) low initially, then increase depth as training progresses. We therefore ask whether LR warmup is necessary if curvature is gated by architecture. As shown in Fig. 6, models equipped with architecture warmup, achieves on par convergence while exceeding the stability of LR warmup.

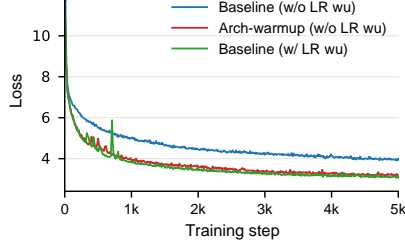

Figure 6: **Replacing LR warmup with Arch-warmup.** A 16-layer (1B) Transformer trained on FineWeb using *architecture warm-up without* learning-rate warmup attains on-par convergence while exhibiting more stable training.

Here, the baseline is trained with a standard, well-tuned learning-rate schedule with warm-up. Concretely, we use commonly adopted hyperparameters for LLaMA-style models: peak learning rate $4 \times 10^{-4}$, weight decay $0.1$, 2000 warm-up steps, and a cosine decay schedule.

The *architecture-warm-up–only* variant is obtained by taking this baseline configuration and removing only the LR warm-up: we set the number of warm-up steps to zero, keep the peak learning rate and decay schedule unchanged, keep all optimizer hyperparameters (including weight decay) fixed, and then enable architecture warm-up.

This suggests architecture warm-up has the potential to *replace* LR warmup in practice, or be combined with it for an even wider stable operating range.

## 5 RELATED WORKS

**Transformer training stability.** Stabilization strategies for transformers span attention- and optimizer-level interventions. On the attention side, *soft-capping* limits logit magnitudes to avoid softmax saturation (Gemma Team, 2024), and $QK$-*normalization* bounds dot-product scales (Henry et al., 2020). On the optimization side, methods adjust or regularize updates (e.g., Adafactor and related stabilizers) (Shazeer & Stern, 2018; Wortsman et al., 2023a). These techniques target proximate causes of loss spikes and are complementary to curvature-based diagnostics that we focus on. **Edge of Stability.** The *Edge of Stability* (EoS) describes the regime where training hovers near the stability boundary, with $\eta\,\lambda_{\max}(H)\approx 2$ for full-batch GD (Cohen et al., 2021; Wang et al., 2022).

Follow-ups generalized the phenomenon to preconditioned/adaptive methods, replacing $H$ by the *preconditioned* Hessian $G_t = P_t^{-1/2} H P_t^{-1/2}$, and analyzed implicit self-stabilization dynamics (Cohen et al., 2022; Damian et al., 2023; Chen & Bruna, 2023). Collectively, these works suggest that practical training often operates close to the spectral stability limit, motivating *online* control of the (preconditioned) top curvature rather than relying solely on fixed hypertuning. Our work aligns with this direction, but extend the analysis from previously explored small scale models to billion parameter transformers proposing an efficient HVP estimation mechanism. **Progressive growing of Transformers.** Prior work increases Transformer capacity during training to cut cost while preserving accuracy: *Progressively Stacking* adds layers stagewise in BERT (Gong et al., 2019; Chen et al., 2020); function-preserving expansions (Net2Net, Network Morphism) enlarge depth/width without changing the realized function (Chen et al., 2016; Wei et al., 2016); and *LiGO* learns growth operators to expand pretrained Transformers with minimal regression (Li et al., 2023). Our focus is stability: we keep the full graph present from initialization and and ensure that depth unlocking preserves curvature. This ties growth to an explicit stability criterion, rather than fixed stage schedules or efficiency alone, and further avoids computational graph surgery.

## 6 Conclusion

We introduce a scalable framework for curvature-aware training of large Transformers. First, we propose an online estimator for the top eigenvalue of the (preconditioned) Hessian that reuses the previous step's eigenvector as a warm start. Under standard smoothness and eigengap assumptions, we prove that the leading eigendirection is slow-moving, which yields rapid geometric convergence of warm-started power iteration. Then, we propose *architecture warm-up*: a function-preserving mechanism that progressively increases the effective depth according to a curvature budget, thereby controlling the growth of effective curvature. Empirically, this combination broadens the range of stable learning rates, reduces loss spikes, and improves training reliability.

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

# 7 APPENDIX

## A PROOFS

### A.1 PROOF FOR THEOREM 1

*Proof.* First, We recall a useful result from Davis–Kahan, sin-$\Theta$ theorem.

> Let $\Sigma, \widehat{\Sigma} \in \mathbb{R}^{p \times p}$ be symmetric, with eigenvalues $\lambda_1 \geq \cdots \geq \lambda_p$ and $\hat{\lambda}_1 \geq \cdots \geq \hat{\lambda}_p$, respectively. Fix $1 \leq r \leq s \leq p$ and assume that
>
> $$\min\{\lambda_{r-1} - \lambda_r, \ \lambda_s - \lambda_{s+1}\} > 0,$$
>
> where we define $\lambda_0 = \infty$ and $\lambda_{p+1} = -\infty$. Let $d = s - r + 1$, and let $V = (v_r, v_{r+1}, \ldots, v_s) \in \mathbb{R}^{p \times d}$ and $\widehat{V} = (\hat{v}_r, \hat{v}_{r+1}, \ldots, \hat{v}_s) \in \mathbb{R}^{p \times d}$ have orthonormal columns satisfying $\Sigma v_j = \lambda_j v_j$ and $\widehat{\Sigma} \hat{v}_j = \hat{\lambda}_j \hat{v}_j$ for $j = r, r+1, \ldots, s$. Then
>
> $$\left\| \sin \Theta(\widehat{V}, V) \right\|_{\mathrm{F}} \leq \frac{2 \min\left(d^{1/2} \|\widehat{\Sigma} - \Sigma\|_{\mathrm{op}}, \ \|\widehat{\Sigma} - \Sigma\|_{\mathrm{F}}\right)}{\min\left(\lambda_{r-1} - \lambda_r, \ \lambda_s - \lambda_{s+1}\right)}. \tag{2}$$

We set $r = s = 1$. Then we have, $\min\left(\lambda_{r-1} - \lambda_r, \ \lambda_s - \lambda_{s+1}\right) = \lambda_1 - \lambda_2$. Further, $\widehat{V} = \hat{v}_1$ and $V = v_1$, $d = 1$.

Substituting $\widehat{\Sigma} = H_{k+1}$ and $\Sigma = H_k$ to above result, and with the Lipschitz condition, we have

$$\sin \varepsilon_k \leq \frac{2 \min\left( \|H_{k+1} - H_k\|_{\mathrm{op}}, \ \|H_{k+1} - H_k\|_{\mathrm{F}}\right)}{\gamma}$$

And we know, $\min\left( \|H_{k+1} - H_k\|_{\mathrm{op}}, \ \|H_{k+1} - H_k\|_{\mathrm{F}}\right) = \|H_{k+1} - H_k\|_{\mathrm{F}}$.

So we have,

$$\sin \varepsilon_k \leq \frac{\|H_{k+1} - H_k\|}{\gamma} \leq \frac{L_H}{\gamma} \|\theta_{k+1} - \theta_k\|. \tag{17}$$

For gradient descent, $\|\theta_{k+1} - \theta_k\| = \eta_k \|\nabla f(\theta_k)\|$. If $\theta_{k+1} = \theta_k - \eta_k g_k$ with $\mathbb{E}[g_k \mid \theta_k] = \nabla f(\theta_k)$ and $\mathbb{E}\|g_k\|^2 \leq G^2$, then $\mathbb{E}\left[ \sin \varepsilon_k \mid \theta_k \right] \leq (L_H/\gamma) \eta_k \mathbb{E}\|g_k\| \leq (L_H/\gamma) \eta_k G$. $\qquad\square$

### A.2 PROOF FOR THEOREM 2

*Proof.* For a symmetric matrix with simple top eigenvalue,

$$\tan \theta_{t+1} = \frac{\|(I - v_1 v_1^\top) A y^{(t)}\|}{|\langle v_1, A y^{(t)} \rangle|} \leq \frac{|\lambda_2|}{|\lambda_1|} \tan \theta_t = \rho \tan \theta_t,$$

hence by induction

$$\tan \theta_t \leq \rho^t \tan \theta_0.$$

Take $A = H_{k+1}$, $v_1 = v_{1,k+1}$, and $y^{(0)} = v_{1,k}$. Then

$$\theta_0 = \angle\left(y^{(0)}, v_{1,k+1}\right) = \angle\left(v_{1,k}, v_{1,k+1}\right) = \varepsilon_k,$$

so with $\rho_{k+1} := \lambda_{2,k+1}/\lambda_{1,k+1} \in [0, 1)$,

$$\tan \theta_t \leq \rho_{k+1}^t \tan \varepsilon_k,$$

Expand $y$ in the $H_{k+1}$-eigenbasis:

$$y^\top H_{k+1} y = \sum_{i \geq 1} \lambda_{i,k+1} \langle y, v_{i,k+1} \rangle^2.$$

Therefore,

$$\lambda_{1,k+1} - y^\top H_{k+1} y = \sum_{i \geq 2} \left(\lambda_{1,k+1} - \lambda_{i,k+1}\right) \langle y, v_{i,k+1} \rangle^2 \leq \left(\lambda_{1,k+1} - \lambda_{2,k+1}\right) \sum_{i \geq 2} \langle y, v_{i,k+1} \rangle^2,$$

and since $\sum_{i \geq 2} \langle y, v_{i,k+1} \rangle^2 = \sin^2 \theta_t$,

$$0 \leq \lambda_{1,k+1} - y^\top H_{k+1} y \leq \left(\lambda_{1,k+1} - \lambda_{2,k+1}\right) \sin^2 \theta_t.$$

Using equation A.2 and $\sin \theta_t \leq \tan \theta_t$ on $[0, \frac{\pi}{2}]$,

$$\lambda_{1,k+1} - y^\top H_{k+1} y \leq \left(\lambda_{1,k+1} - \lambda_{2,k+1}\right) \rho_{k+1}^{2t} \tan^2 \varepsilon_k = (1 - \rho_{k+1}) \lambda_{1,k+1} \rho_{k+1}^{2t} \tan^2 \varepsilon_k,$$

which is equation 8.

By Theorem 1,

$$\tan \varepsilon_k \leq \sin \varepsilon_k \leq \frac{L_H}{\gamma} \|\theta_{k+1} - \theta_k\| \quad \Rightarrow \quad \tan \theta_t \leq \rho_{k+1}^t \frac{L_H}{\gamma} \|\theta_{k+1} - \theta_k\|.$$

To ensure $\theta_t \leq \delta \in (0, \frac{\pi}{2})$, it suffices that

$$\rho_{k+1}^t \frac{L_H}{\gamma} \|\theta_{k+1} - \theta_k\| \leq \tan \delta.$$

Since $0 < \rho_{k+1} < 1$, taking logs yields

$$t \geq \frac{\log\left(\frac{L_H}{\gamma} \|\theta_{k+1} - \theta_k\|\right) - \log(\tan \delta)}{\log(1/\rho_{k+1})},$$

.

Let $x \sim \mathrm{Unif}(\mathbb{S}^{d-1})$. With high probability,

$$\langle x, v_{1,k+1} \rangle^2 = \Theta(1/d) \quad \Rightarrow \quad \tan \angle(x, v_{1,k+1}) = \Theta(\sqrt{d}).$$

Thus $\tan \theta_t \leq \rho_{k+1}^t \Theta(\sqrt{d}) \leq \tan \delta$ implies

$$t_{\mathrm{rand}} \gtrsim \frac{\frac{1}{2} \log d - \log(\tan \delta)}{\log(1/\rho_{k+1})},$$

and subtracting the warm-start bound gives equation 9. $\qquad \square$

## A.3 IMPLEMENTATION SKETCH OF WARM-START POWER ITERATION

---

**Algorithm 1** Warm-Start HVP Power Iteration for $\lambda_{\max}(H(\theta))$

---

1: **input:** parameters $\theta$; optional warm vector $y_{\mathrm{warm}}$ with $\|y_{\mathrm{warm}}\|_2 = 1$; optional previous estimate $\hat{\lambda}_{\mathrm{warm}}$; tolerance $\varepsilon$
2: **initialize:**
3: $\quad y^{(0)} \leftarrow \begin{cases} y_{\mathrm{warm}}, & \text{if provided} \\ \text{randn unit vector}, & \text{otherwise} \end{cases}$
4: $\quad$ **(early exit check)** If $\hat{\lambda}_{\mathrm{warm}}$ provided, optionally compute residual $r^{(0)} \leftarrow \|(H(\theta) - \hat{\lambda}_{\mathrm{warm}}I)y^{(0)}\|_2$ (1 HVP). If $r^{(0)} \leq \varepsilon |\hat{\lambda}_{\mathrm{warm}}|$, **return** $(\hat{\lambda}_{\mathrm{warm}}, y^{(0)})$.
5: **for** $t = 0, 1, 2, \ldots$ **do**
6: $\quad z^{(t+1)} \leftarrow H(\theta) y^{(t)}$ {Pearlmutter HVP}
7: $\quad$ **guard:** If $\|z^{(t+1)}\|_2 = 0$ (numerical underflow), reinitialize $y^{(t)}$ to a fresh random unit vector and **continue**.
8: $\quad y^{(t+1)} \leftarrow z^{(t+1)}/\|z^{(t+1)}\|_2$ {power update}
9: $\quad \hat{\lambda}^{(t+1)} \leftarrow \langle y^{(t+1)}, H(\theta) y^{(t+1)} \rangle$ {one extra HVP or reuse if cached}
10: $\quad$ **warm-start stabilization (optional):**
11: $\quad\quad$ *(i) Momentum mix:* if $t = 0$ and $y_{\mathrm{warm}}$ provided, set $y^{(1)} \leftarrow$ normalize$\left(\alpha y^{(1)} + (1-\alpha) y_{\mathrm{warm}}\right)$ with small $(1-\alpha)$ (e.g., 0.1).
12: $\quad\quad$ *(ii) Cosine guard:* if $t = 0$ and $|\langle y^{(1)}, y_{\mathrm{warm}} \rangle| < \tau$ (e.g., $\tau$=0.1), replace $y^{(1)} \leftarrow y_{\mathrm{warm}}$.
13: $\quad$ **stopping test:** If $\|(H(\theta) - \hat{\lambda}^{(t+1)}I)y^{(t+1)}\|_2 \leq \varepsilon |\hat{\lambda}^{(t+1)}|$, **return** $(\hat{\lambda}^{(t+1)}, y^{(t+1)})$.
14: **end for**

---

Note that the proposed warm-start strategy is particularly well suited for large neural networks where HVPs are expensive and frequent monitoring of curvature is desirable. By amortizing eigenvector estimation across training steps, we enable efficient online tracking of sharpness without compromising accuracy. To the best of our knowledge, this continuation-style use of power iteration has not been previously explored in the deep learning literature, despite being a standard idea in numerical linear algebra. Our theoretical guarantees and empirical results demonstrate its clear superiority over random initialization for curvature estimation in neural network training.

## B   DEPTH, LIPSCHITZ GROWTH, AND CURVATURE

In this section, we discuss the connection between the depth and the stability margin with formal results.

**Setup and notation.**   Let $g_\theta = \Phi_L \circ \cdots \circ \Phi_1$ with residual blocks $\Phi_\ell(x) = x + B_\ell(x)$. Write $J_\ell(x) = I + \partial B_\ell(x)$, $J_x g_\theta$ for the input Jacobian, and $x_0 = x$, $x_\ell = \Phi_\ell(x_{\ell-1})$. Norms are operator (spectral) norms.

**Lemma 1** (Depth–Lipschitz bound).   *For all $x$,*

$$\|J_x g_\theta(x)\| \;\leq\; \prod_{\ell=1}^{L} \|J_\ell(x_{\ell-1})\| \;\leq\; \exp\Big(\sum_{\ell=1}^{L} \|\partial B_\ell(x_{\ell-1})\|\Big).$$

*Proof.* By the chain rule, $J_x g_\theta(x) = J_L(x_{L-1}) \cdots J_1(x_0)$. Submultiplicativity gives $\|J_x g_\theta(x)\| \leq \prod_\ell \|J_\ell(x_{\ell-1})\|$. Next use $\|I + A\| \leq 1 + \|A\| \leq \exp(\|A\|)$ to obtain the exponential bound.  □

**Lemma 2** (Parameter sensitivity growth).   *Let $J_\theta g_\theta(x)$ denote the Jacobian of $g_\theta$ w.r.t. parameters. Then $\|J_\theta g_\theta(x)\| \leq C \prod_{\ell=1}^{L} \|J_\ell(x_{\ell-1})\|$ for some constant $C$ depending on block parametrization (e.g., linear maps and elementwise activations yield $C = 1$ up to dimension factors). In particular, $J_\theta g_\theta$ inherits the multiplicative growth in Lemma 1.*

*Proof.* Differentiate the composition with respect to block parameters; each term contains a product of input Jacobians $J_k$ before and after the block where parameters appear. Bounding each product by $\prod_\ell \|J_\ell\|$ gives the stated inequality (constants collect per-block linear maps).  □

**Proposition 1** (Curvature bound via Gauss–Newton).   *Assume $\ell$ is twice differentiable in $z = g_\theta(x)$ with $\lambda_{\max}(\nabla_z^2 \ell) \leq L_\ell$. Then*

$$\lambda_{\max}\big(\nabla_\theta^2 \mathcal{L}(\theta)\big) \;\leq\; L_\ell \big(\sup_x \|J_\theta g_\theta(x)\|\big)^2.$$

*Proof.* For each $(x, y)$, the Gauss–Newton term is $J_\theta g_\theta(x)^\top \nabla_z^2 \ell\, J_\theta g_\theta(x) \preceq L_\ell\, J_\theta g_\theta(x)^\top J_\theta g_\theta(x)$, hence the stated bound after taking expectation and the maximum eigenvalue.  □

**Corollary 1** (Preconditioned curvature).   *Let $P_t$ be SPD diagonal with $m_t I \preceq P_t \preceq M_t I$. Then for $G_t = P_t^{-1/2} H P_t^{-1/2}$,*

$$\frac{1}{M_t}\,\lambda_{\max}(H) \;\leq\; \lambda_{\max}(G_t) \;\leq\; \frac{1}{m_t}\,\lambda_{\max}(H).$$

*Hence the depth dependence of $\lambda_{\max}(G_t)$ mirrors that of $\lambda_{\max}(H)$ up to constant factors $(m_t, M_t)$.*

*Proof.* For SPD $P_t$, Rayleigh quotients give $\lambda_{\max}(G_t) = \max_{\|v\|=1} v^\top P_t^{-1/2} H P_t^{-1/2} v = \max_{\|u\|_{P_t}=1} u^\top H u$, where $\|u\|_{P_t}^2 = u^\top P_t u$. Using $m_t \|u\|^2 \leq \|u\|_{P_t}^2 \leq M_t \|u\|^2$ yields the bounds.  □

**Corollary 2** (Stability margin scales as $1/\lambda_{\max}$).   *For a quadratic model of the local dynamics, gradient descent is stable if $\eta < 2/\lambda_{\max}(H)$; with momentum/Adam, the admissible region for $(\eta, \beta)$ scales as $O(1/\lambda_{\max}(G_t))$. Combining Lemma 1, Lemma 2, and Proposition 1 shows that increasing depth $L$ raises $\lambda_{\max}(H)$ (and $\lambda_{\max}(G_t)$), thereby shrinking the stability margin.*

## C    SELF-STABILIZATION OF CURVATURE DURING TRAINING

### C.1    CURVATURE AS TRAINING PROGRESSES

We track the raw curvature $\lambda_{\max}(H(\theta_t))$, the *effective* (preconditioned) curvature $\lambda_{\max}(G_t)$, and the preconditioner inverse $P_t^{-1}$ for a 16-layer (1B) model. As shown in Fig. 7, after an initial transient the dynamics enter a regime where $\lambda_{\max}(G_t)$ oscillates within a narrow band, consistent with the edge-of-stability picture. This self-stabilization supports increasing depth later in training, once curvature has settled. Notably, $P_t^{-1}$ continues to grow over time, indicating a steadily strengthening preconditioning effect from the optimizer.

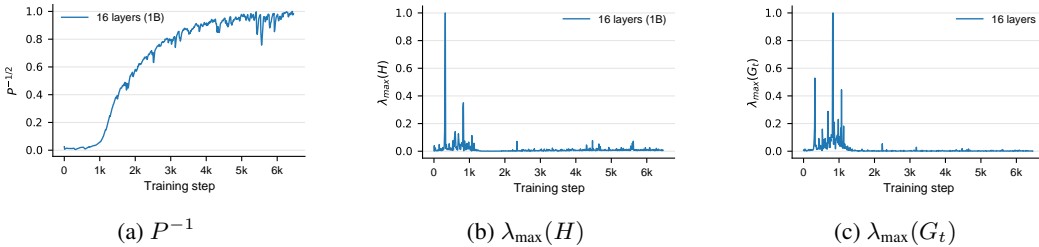

|  |  |  |
|---|---|---|
| (a) $P^{-1}$ | (b) $\lambda_{\max}(H)$ | (c) $\lambda_{\max}(G_t)$ |

Figure 7: **Self stabilization of transformers.** After an initial transient period, both the raw curvature and the preconditioned curvature settles down to a stable, lower band. This supports the effectiveness of increasing the depth later in the training. Interestingly, $P^{-1}$ keeps growing as the training progresses (shown values are normalized).

### C.2    SELF STABILIZATION OF CURVATURE DURING TRAINING

We observe a self-stabilization effect in Transformers (see Fig. 8): when curvature rises, the optimizer's preconditioner counteracts it. In Adam-like methods with $P_t = \mathrm{diag}(\sqrt{v_t} + \varepsilon)$ and $v_{t+1} = \beta_2 v_t + (1 - \beta_2)g_t^{\odot 2}$, larger curvature typically coincides with larger gradients $g_t$, which increases $v_t$ and thus $P_t$. Since the effective curvature is $G_t = P_t^{-1/2} H P_t^{-1/2}$, a larger $P_t$ (smaller $P_t^{-1}$) reduces Rayleigh quotients and damps $\lambda_{\max}(G_t)$, yielding a stabilizing feedback. See below for a formal discussion.

**Self-stabilization via adaptive preconditioning (and its limits).**    Let $H_t := H(\theta_t)$ and consider an Adam-like update $\theta_{t+1} = \theta_t - \eta\, M_t^{-1}\, \widehat{m}_t$ with diagonal preconditioner $M_t = \mathrm{diag}(\sqrt{v_t} + \varepsilon)$, where

$$v_{t+1} = \beta_2 v_t + (1 - \beta_2)\, g_t^{\odot 2}, \qquad g_t = \nabla\mathcal{L}(\theta_t).$$

The *effective* (preconditioned) curvature experienced by the optimizer is

$$G_t \;=\; M_t^{-1/2}\, H_t\, M_t^{-1/2}, \qquad \lambda_{\max}(G_t) \;=\; \max_{\|x\|=1} x^\top M_t^{-1/2} H_t M_t^{-1/2} x.$$

When curvature or gradient energy surges, $g_t^{\odot 2}$ increases and (after EMA smoothing) $M_{t+1}\uparrow$; consequently $M_{t+1}^{-1/2}\downarrow$ and *all* Rayleigh quotients of $G_{t+1}$ decrease. A simple bound follows from the Rayleigh quotient and diagonal ordering:

$$\lambda_{\max}(G_{t+1}) \;\leq\; \|M_{t+1}^{-1/2}\|_2^2\, \lambda_{\max}(H_{t+1}) \;=\; \frac{\lambda_{\max}(H_{t+1})}{\min_i \left(\sqrt{v_{t+1,i}} + \varepsilon\right)^2}.$$

Thus, as $v_{t+1}$ grows, the preconditioner shrinks the effective curvature, exhibiting an *implicit self-stabilization* that nudges the product $\eta\, \lambda_{\max}(G_t)$ toward the stability band (the EoS).

*However, this is not sufficient to ensure stable convergence.* Despite this automatic balancing, there are three failure modes: (i) **lag:** $v_t$ reacts on a time scale $\sim \frac{1}{1-\beta_2}$ steps, so sharp, step-scale spikes in $H_t$ can push $\eta\, \lambda_{\max}(G_t)$ beyond the boundary before $M_t$ catches up; (ii) **anisotropy:** $M_t$ is diagonal, whereas $H_t$ can be highly anisotropic; a coordinate-wise preconditioner cannot instantly

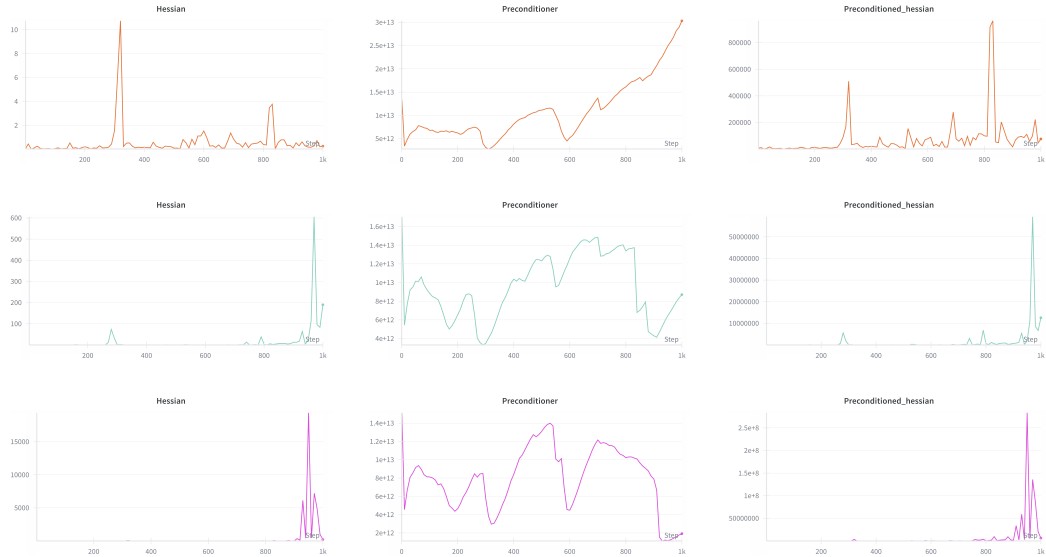

Figure 8: **Self-stabilization in Transformers.** Close-ups over particular training windows of second- and first-order statistics for 8-, 16-, and 32-layer models. Each row (left→right) shows $\lambda_{\max}(H)$, $P^{-1}$, and $\lambda_{\max}(G_t)$. When the Hessian spikes, the preconditioner dips, reducing $\lambda_{\max}(G_t)$ and partially stabilizing the effective curvature; however, this feedback is not always sufficient to keep $\lambda_{\max}(G_t)$ within the stability threshold.

suppress a sharp *direction* that is a dense combination of coordinates; (iii) **coupling with momentum:** with $\beta_1 > 0$, a large $m_t$ can overshoot even if $M_t$ starts to grow, transiently amplifying the update.

In summary, adaptive methods do provide a *reactive* stabilizer, $M_t^{-1}$ tends to drop as curvature rises, reducing $\lambda_{\max}(G_t)$, but this mechanism is imperfect under fast spikes, strong anisotropy, or momentum coupling. Our *architecture warm-up* complements this by acting *proactively* on $H_t$ itself: by keeping depth (and hence the operator norm of intermediate Jacobians) low early and unlocking blocks later in training, we keep the system inside the stability envelope even when the optimizer's preconditioner has not yet adapted.

## D  VALIDATION AT COMPUTE OPTIMAL

Since *QK-Norm* was the strongest baseline in our shorter FineWeb runs, we benchmark it at the *Chinchilla* compute-optimal setting (Hoffmann et al., 2022): a 1B-parameter, 16-layer model trained for 25B tokens on FineWeb. This experiment tests whether early-phase gains from Arch-Warmup persist at compute-optimal budget. As Table 2 shows, Arch-Warmup outperforms QK-Norm at this scale, indicating stronger convergence and stability that carry through to the compute-optimal regime

## E  SENSITIVITY TO THE WARM UP SCHEDULE

.

By default, we keep the model at half depth until learning-rate warmup completes, then unlock the remaining layers in four groups spaced by 500 training iterations to reach full depth. Performance, however, is not sensitive to this spacing: Fig. 10 shows that even with *zero* spacing (i.e., unlocking all remaining layers at once at peak learning rate), results are similar. The reason is that newly enabled blocks are *zero-initialized*, so their contribution to the Jacobian/Hessian—and thus the total (preconditioned) curvature—grows gradually from the half-depth baseline. The model therefore

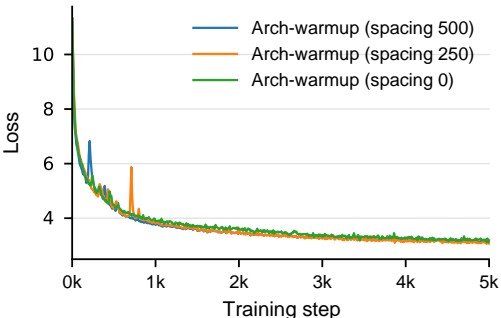

Figure 9: Convergence with different architecture warm-up schedules. We observe that our method is not highly sensitive to the schedule.

Table 2: Validation perplexity (PPL) on FineWeb at Chinchilla compute-optimal (1B, 16 layers, 25B tokens). Lower is better.

| Method | Val PPL ↓ |
|---|---|
| QK-Norm | 20.28 |
| Softcap | 32.44 |
| Arch-Warmup | **18.35** |

never encounters a sudden "full-depth" curvature jump at the unlock step; instead, capacity and curvature are realized progressively as those weights move away from zero.

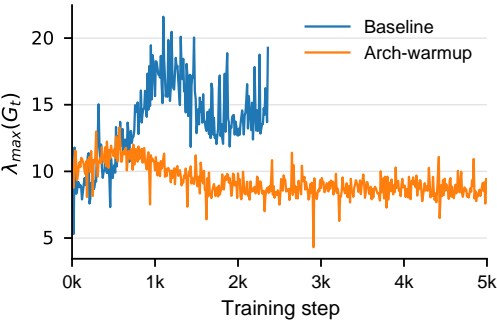

Figure 10: Log-scale plot for the evolution of $\lambda_{\max}(G_t)$ with a peak learning rate of $8 \times 10^{-3}$.

