# OpenReview forum: "Taming Curvature: Architecture Warm-up for Stable Transformer Training"
_ICLR.cc/2026/Conference — ICLR 2026 Poster_

### Official Review · Reviewer_KvM5 · 2025-10-27

**Soundness:** 3
**Presentation:** 3
**Contribution:** 2
**Rating:** 6
**Confidence:** 4

**Summary:**

In this work the authors develop a technique called "architecture warmup" to control curvature. In short, they initialize many layers to 0, and slowly bring them online into training. Due to the residual connections the initial reduced network trains well, and the new layers are brought online continuously. The authors use a warmstarted power iteration to measure sharpness and show that this method initializes at lower sharpness, and ends the procedure at higher sharpness. They show that this can allow for training at larger initial learning rates, and that it can obtain similar training curves to traditional warmup.

**Strengths:**

The idea of architecture warmup is simple and, to my knowledge, novel; I admit that I am not familiar with the literature on growing networks so I will defer to other reviewers on this point. Overall the simplicity of the idea and its general compatibility with common training schemes is a big strength. The experiments seem sound to the best of my ability to understand them.

The idea of warmstarting the power iterations is also simple yet effective, and I appreciate the authors for running experiments demonstrating its utility as a measurement tool.

**Weaknesses:**

The main question is whether or not the proposed architecture warmup provides benefits that are better than learning rate warmup. Of particular interest is the case of the large transformer models where depth and width are often scaled simultaneously. Indeed the resulting learning curves are quite similar to transformers trained with warmup.

There is a broader point that simply allowing learning with higher learning rates is not in and of itself a win; for example if the gradient norms are smaller than the effect of a larger learning rate can be cancelled out. This suggests that a deeper investigation of the method is needed to understand why it might be useful.

One potential issue with the curvature measurements: typically experimentalists do power iteration on a small batch of data to compute an estimate of the large Hessian eigenvalues. However, this does not give an unbiased estimator of the full batch Hessian, and also may give a noisy estimator of the batch-Hessian maximum eigenvalue. This makes interpreting the curvature measurements in the paper more difficult.

**Questions:**

Do the minibatch curvature measurements well-reflect the full batch curvature statistics? Can evidence be provided using e.g. HVP aggregation over multiple batches?

Is there any way to test whether or not networks trained with architecture warmup have benefits in the fine-tuning setting, where the curvature of the pretrained model may be more predictive of downstream success?

Suggestion: for the warm started curvature measurement, it might be better to spend less time justifying the procedure, and more time describing the experiments.

---

> ### Author Response · Authors · 2025-11-20
>
> We thank the reviewer for the positive feedback and valuable comments. Please find our response below.
>
> **1. The main question is whether or not the proposed architecture warmup provides benefits that are better than learning rate warmup. Of particular interest is the case of the large transformer models where depth and width are often scaled simultaneously. Indeed the resulting learning curves are quite similar to transformers trained with warmup.**
>
> We agree that this is a central question, especially in the large-scale transformer regime where depth and width are scaled together and learning-rate (LR) warmup is already standard practice.
>
> In our main experiments, all baselines use a conventional LR warmup schedule; architecture warmup is applied in addition to LR warmup, not instead of it. Thus, when learning curves look similar at a coarse scale, they should be interpreted as “LR warmup vs. LR warmup + architecture warmup,” rather than as architecture warmup used in isolation. In this setting, our curvature measurements show that, even with well-tuned LR warmup, deep models tend to enter or hover near the edge-of-stability regime at reasonable peak learning rates, leading to sharpness spikes and, in some cases, divergence. Architecture warmup mitigates this by starting from a shallower, better-conditioned model and only gradually increasing depth, which reduces curvature spikes and improves robustness for the same LR schedule.
>
> Conceptually, LR warmup and architecture warmup act on different aspects of the dynamics: LR warmup modulates the effective step size, while architecture warmup modulates the effective depth (and thus curvature) over time. This makes them complementary rather than redundant.
>
> To directly address the question of replacement, we also include an exploratory experiment ( Fig. 6) where we remove LR warmup and rely solely on architecture warmup. In that setup, training remains stable and achieves competitive performance, suggesting that architecture warmup can, in some regimes, partially substitute for LR warmup. We are careful to present this as preliminary evidence (based on a single configuration) rather than a general claim of universal replacement. Our main message is therefore twofold: (i) architecture warmup provides additional stability and robustness on top of a strong LR-warmup baseline, and (ii) it shows promising potential to reduce reliance on LR warmup in some settings, which we view as an interesting direction for future large-scale studies.

---

> > ### Comment · Reviewer_KvM5 · 2025-11-20
> > **Response 1**
> >
> > I thank the authors for their comments, I will respond point by point.
> >
> > Thank you for the clarification on the results here; in particular I appreciate that the authors took great pains not to overclaim their results!

---

> ### Author Response · Authors · 2025-11-20
>
> **2. There is a broader point that simply allowing learning with higher learning rates is not in and of itself a win; for example if the gradient norms are smaller than the effect of a larger learning rate can be cancelled out. This suggests that a deeper investigation of the method is needed to understand why it might be useful.**
>
> This is an important point, and we agree that “being able to use a higher learning rate” is not, by itself, evidence that a method is better. In particular, if gradient norms are reduced at the same time, a larger nominal learning rate can be effectively cancelled out.
>
> Note, however, that for adaptive optimizers such as Adam, the update is roughly of the form
>
> $\Delta \theta_t \propto \eta \frac{g_t}{\sqrt{v_t}}$
>
> so that simple rescaling of the gradient norm is partly absorbed by the normalization through the second-order moment. Also, please note that recent works have also shown that operating in a higher–learning-rate regime can be beneficial for convergence and generalization when the convergence is stable way [1].
>
> On the other hand, a smaller learning rate can indeed keep the effective curvature–step-size product  $\lambda_{\max}(H_t)\eta$ further from the edge-of-stability boundary and thereby improve stability. The question we emphasize is whether this is an efficient way to achieve stability. In principle, one can always choose η\etaη small enough that the sharpness never approaches the threshold $2/\eta$, but such step sizes are often suboptimal in terms of training speed and cost (sometimes dramatically so) especially in today’s extremely expensive LLM training regimes. In practice, recent large-scale model reports consistently show that, at reasonable step sizes and learning-rate schedules, billion-parameter–scale models enter unstable regions and even diverge, a phenomenon we connect to the edge-of-stability regime using our mathematical tools.
>
> Conceptually, our goal is not to argue that curvature cannot be controlled by other means (e.g., more conservative learning rates), but rather to show that (i) curvature is a key mechanism underlying the observed training behaviour, and (ii) architecture warm-up provides a complementary way to steer optimization into more stable regions without resorting to overly conservative learning rates. To support this, we go beyond simply reporting learning curves: we analyse sharpness trajectories, curvature estimates, and instability/divergence statistics, precisely as an initial step toward the “deeper investigation” the reviewer is calling for. We view our current study as a step in that direction, and we agree that extending this analysis to a broader range of models and settings is an important avenue for future work.
>
> [1] - Methods of improving LLM stability.
>
> **3.  Do the minibatch curvature measurements well-reflect the full batch curvature statistics? Can evidence be provided using e.g. HVP aggregation over multiple batches?**
>
> This is an excellent point by the reviewer. We agree that, in general, stochastic gradient descent can exhibit different dynamics near the edge of stability compared to full-batch gradient descent. However, the literature also shows that this discrepancy shrinks rapidly as the batch size increases (typically once it exceeds ≈128), so that large-batch SGD is well-approximated by the full-batch theory.
> Concretely, [1, Fig. 4] shows that for MNIST, once the batch size exceeds 128, the observed dynamics quickly converge to the full-batch predictions. Similarly, [2, Fig. 24] demonstrates that sharpness dynamics are accurately predicted by the full-batch theory for batch sizes larger than 128, for both MSE and cross-entropy losses.
>
> Typically, LLM training is performed with very large batch sizes (up to several thousand tokens per step. In our experiments we use a batch size of 1024. Moreover, for decoder-only Transformers with cross-entropy loss, the gradient at each step is an average over all tokens in all sequences in the batch. Thus, a batch of 1024 sequences of length 1024 corresponds to gradients averaged over ≈10⁶ token-level contributions, an effective batch size far beyond the $\approx 128$ regime where SGD dynamics are already well approximated by GD in prior work. For this reason, although SGD and GD can differ in principle, in our regime it is reasonable to use full-batch curvature intuitions to interpret and predict the observed behavior of architecture warm-up, which is what our analysis aims to do. In our experiments we also observe that
>
> $\log \bigl(\lambda_{\max}(G_t)\bigr) \approx 8.5$
>
> So that, at the peak learning rate $\eta = 0.008$
>
> $\lambda_{\max}(G_t)\eta \approx \exp(8.5) \cdot 0.008 \approx  39.3 \approx 38$
>
> which is the stability threshold for full batch Adam.
>
>
> [1] - High dimensional analysis reveals conservative sharpening and a stochastic edge of stability
>
> [2] -  Gradient Descent on Neural Networks Typically Occurs at the Edge of Stability

---

> > ### Comment · Reviewer_KvM5 · 2025-11-20
> > **Response 2**
> >
> > With regards to the questions about learning rate:
> >
> > To me, even with an optimizer like Adam, there are three questions with regards to the efficacy of larger learning rate:
> >
> > 1. What is the update size relative to the parameter norms? ($\eta||u||$ vs $||\theta||$).
> > 2. Does a larger learning rate lead to better decrease in loss? ($\eta u\cdot g$)
> > 3. Does a larger learning rate lead to more change in model outputs for a single step? ($\eta ||J u||$)
> >
> > Any method which increases the learning rate but does not affect these quantities is not really improving learning dynamics, even with a normalized optimizer like AdamW. So to make my comments more precise, my question is: do quantities like this improve with your methodology?
> >
> > With regards to the minibatch curvature vs the full batch curvature: the critical batch size depends quite a bit on the dataset properties. For example, in the quoted reference [1] the training procedure was on a subset of 2500 data points, so the batch fraction (batch size/dataset size) was ~ 5%. Most LLM training has a much smaller batch fraction. It is not clear if the relevant quantity is raw number of datapoints, or fraction of dataset; the details likely depend on the total scale of the data as well as the dataset. Another potential issue is the large number of "classes" (that is, vocab size) in language modeling.
> >
> > That being said, the observation of the consistency of the eigenvalues with the edge of stability is a compelling point. Still, it would be more convincing if evidence could be provided that using 2x or 3x the batch size to compute the largest eigenvalue maintains consistency (or, alternatively, that smaller batch sizes also show the same maximum eigenvalue).

---

> ### Author Response · Authors · 2025-11-20
>
> **4. Is there any way to test whether or not networks trained with architecture warmup have benefits in the fine-tuning setting, where the curvature of the pretrained model may be more predictive of downstream success?**
>
> This is a very interesting question, and we agree that in the fine-tuning regime the curvature of the pretrained model may play a more direct role.
>
> In the current work, however, our primary goal is not to end up with a model that has systematically smaller curvature at the end of training, but rather to keep the effective curvature within the stability margin during the phase where the learning rate is highest and the stability threshold is lowest. In other words, architecture warm-up is designed to stabilize the training trajectory in the large-LR, high-instability regime, not to change the nature of the final solution once both methods have converged to similar validation loss.
>
> For that reason, if two models with the same architecture and data are trained (with and without architecture warm-up) to essentially the same final validation perplexity under comparable hyperparameters, our expectation is that their downstream and fine-tuning performance will also be very similar. This is consistent with the standard practice in large-scale language modeling, where validation loss is used as the main proxy for overall model quality, and where downstream performance typically correlates strongly with upstream loss.
>
> That said, your suggestion does point to a concrete and interesting follow-up: one could pretrain models with and without architecture warm-up to matched loss, then (i) measure curvature/flatness of the resulting checkpoints and (ii) compare their performance and stability in a controlled fine-tuning suite. Such experiments are beyond our current scope and compute budget, but we agree they would be a valuable next step to more directly probe whether curvature differences at the end of pretraining carry over to fine-tuning, and we will clarify this in the revision as a direction for future work.
>
> **5. Suggestion: for the warm started curvature measurement, it might be better to spend less time justifying the procedure, and more time describing the experiments.**
>
> Thank you for this suggestion. We agree that the current presentation may overemphasize the justification of the warm-started curvature procedure at the expense of experimental detail. In the camera ready version, we will streamline the theoretical/justification part and use the saved space to better describe the experimental setup and findings.

---

> ### Author Response · Authors · 2025-11-24
> **Response to further comments.**
>
> We thank the reviewer for clarifying the question and for the intriguing follow up questions. In order to address the reviewer's concerns, we have added plots where,
>
> 1. It shows that increasing the learning rate increases the convergence rate, but after a certain threshold, the training diverges due to instability. This is precisely what architecture warmup aims to handle.
>
> 2. We now report largest-eigenvalue estimates for multiple batch sizes in our setting. In particular, we compute the largest eigenvalue for preconditioned Hessians using batch sizes 1024, 2048, and 4096 and plot these on a logarithmic scale. Across these larger batch sizes, the eigenvalue trajectories essentially overlap within the noise level, especially in the edge-of-stability regime that we focus on. This indicates that, in our configuration, the edge-of-stability dynamics we measure are robust to a 2x, 4x change in batch size, which is in line with the reviewer’s suggestion to verify that the largest eigenvalue estimates remain consistent when using larger batches.
>
> Please find the plots in supplementary at *torchtitan-archwarmup-arch-warmup-clean\plots*
>
> Regarding the citation we provided, we agree that the critical batch size can depend strongly on dataset properties, and that the batch fraction and other factors, such as the large number of classes (vocabulary size), can play an important role in language modeling. We did not intend our earlier citation to be read as a rigorous proof for the LLM regime. Rather, it was meant as supporting evidence that, beyond a certain scale, SGD dynamics can closely track full-batch predictions. Further rigorous treatment for this problem would be an interesting future research direction.

---

> > ### Comment · Reviewer_KvM5 · 2025-11-26
> > **Thank you for the additional experiments**
> >
> > The additional experiments address my concerns about the curvature measurements; thanks for including them!

---

### Official Review · Reviewer_4tpE · 2025-10-29

**Soundness:** 3
**Presentation:** 3
**Contribution:** 3
**Rating:** 6
**Confidence:** 4

**Summary:**

This paper addresses training instabilities in large-scale Transformers through curvature control. The authors propose: (1) an efficient online estimator for the largest (preconditioned) Hessian eigenvalue using warm-started power iteration, reducing computational cost from >20 to <5 HVPs; (2) an "architecture warm-up" strategy that progressively increases network depth to maintain the stability criterion η·λmax(Gt) within safe bounds. Experiments on models up to 3B parameters demonstrate improved stability compared to QK-Norm, Softcap, and QK-Clip.

**Strengths:**

1.	Solid Theory: Provides formal analysis connecting Edge of Stability theory with practical deep learning, with non-trivial theoretical bounds
2.	Practical and Simple: Easy integration into existing pipelines without architectural surgery or complex hyperparameter tuning
3.	Comprehensive Experiments: Multiple datasets (FineWeb, DCLM, OLMo-Mix), various model sizes (640M-3B), and informative ablations
4.	Empirical Validation: Figures 1-2 effectively validate theoretical predictions with proper error bars
5.	Reproducible: Sufficient implementation details with publicly available datasets

**Weaknesses:**

1.	Limited Scale: Maximum 3B parameters tested; no validation on models >10B where instabilities are most severe. The "billion-parameter scale feasibility" claim needs larger-scale demonstration.

2.	Lack of Statistical Rigor: Most experiments show single runs without error bars or significance testing. Table 1 improvements need statistical validation.

3.	Limited Evaluation: Only validation perplexity reported; no downstream task performance to verify that stability improvements translate to better capabilities.

**Questions:**

1.	Computational Cost: Can you provide concrete wall-clock time measurements and memory overhead for 1B and 3B models? What percentage overhead compared to standard training?
2.	Gradient Clipping: How does your method compare against well-tuned gradient clipping? Can they be combined?
3.	Initial Depth Selection: How sensitive are results to starting depth (1/3 vs 1/2 vs 2/3)? Can you provide a principled selection rule?
4.	HVP Iterations: How does the required number of HVPs vary across training stages and model sizes? Should this be adjusted dynamically?
5.	RMSNorm Initialization: What specifically goes wrong with RMSNorm from zero initialization? Did you try alternatives?
6.	Failure Analysis: In Table 1, can you analyze the curvature behavior for methods that diverged (*)?
7.	Compute-Optimal: Can you provide compute-optimal comparisons with all baselines (not just QK-Norm)?
8.	LR Warm-up Replacement: Is Figure 6's result generalizable across model sizes and datasets, or specific to this setting?

---

> ### Author Response · Authors · 2025-11-20
>
> We are grateful for the valuable comments and suggestions. Please find our answers below.
>
> **1. Limited Scale: Maximum 3B parameters tested; no validation on models >10B where instabilities are most severe. The "billion-parameter scale feasibility" claim needs larger-scale demonstration.**
>
> We agree that our empirical validation is limited to models of up to 3B parameters. Due to compute constraints, we are currently unable to run experiments at >10B scale. That said, the algorithm is explicitly designed to have low overhead and to integrate with standard training pipelines (e.g., it does not require architectural changes, custom optimizers, or extra forward passes), and we see no fundamental barrier to applying it to larger models and clusters. We therefore view our results as a first demonstration of feasibility at the billion-parameter scale and as a foundation for future work that can evaluate the method on >10B models, where we agree the question is especially important.
>
> **2. Lack of Statistical Rigor: Most experiments show single runs without error bars or significance testing. Table 1 improvements need statistical validation.**
>
> Thank you for the suggestion. As per the reviewer’s request, we have now added error bars for three runs with three different random seeds.
>
> | Dataset  | Baseline | QK-Norm | QK-Clip | Softcap | Arch-Warmup |
> |---------|---------:|--------:|--------:|--------:|------------:|
> | FineWeb | *        | 49.88 $\pm$ 0.003  | *       | 51.41 $\pm$ 0.002  | **25.02 $\pm$ 0.001**   |
> | DCLM    | 165.62 $\pm$ 0.04   | 61.57 $\pm$ 0.012   | *       | 43.38 $\pm$ 0.03   | **22.64 $\pm$ 0.002**   |
> | OLMo-Mix| *        | *       | *       | *       | **18.54 $\pm$ 0.001**   |
>
> **3. Limited Evaluation: Only validation perplexity reported; no downstream task performance to verify that stability improvements translate to better capabilities.**
>
> Following the reviewer’s suggestion, we have included the results of several benchmarks below.
>
> | Task      | Baseline | QK-Norm | QK-Clip | Softcap | Arch-warmup |
> |-----------|---------:|--------:|--------:|--------:|------------:|
> | ARC Easy  | *        | 0.2511  | *       | 0.2531  | **0.4211**  |
> | HellaSwag | *        | 0.2557  | *       | 0.2341  | **0.3543**  |
> | SciQ      | *        | 0.2745  | *       | 0.2773  | **0.5891**  |
> | PIQA      | *        | 0.4714  | *       | 0.4999  | **0.5842**  |
>
> Further, although we appreciate the reviewer’s concern regarding downstream tasks, we would also like to point out that a large body of work supports using perplexity  as the main proxy for downstream capability. Specifically, [1] fit a power law relating validation perplexity to average downstream top-1 error across a suite of tasks, demonstrating that downstream performance can be reliably predicted from upstream perplexity in realistic LLM settings. More recently, [2]  study “loss-to-loss” scaling laws and show that validation and test losses across datasets obey consistent power laws, and that different architectures with similar losses tend to share similar downstream scaling behaviour. [3] further exploit strong perplexity-benchmark correlations to drive data selection, again relying on the fact that improvements in perplexity on relevant validation sets are predictive of improvements on downstream evaluations.
>
> In our setting, we compare models with the same architecture, data, optimizer, and compute budget, differing only in the presence or absence of architecture warm-up. Under these controlled conditions, improvements in validation perplexity and the reduction in unstable/divergent runs are precisely the effects we care about. They indicate that the method makes it easier to reach better solutions in the same training regime, and prior work strongly suggests that such improvements should translate to at least comparable, and typically better, downstream performance. However, we will add the above results to the revised version as per the reviewer’s suggestion. Thank you.
>
> [1] - Language Models Scale Reliably with Over-Training and on Downstream Tasks
>
> [2] - LLMs on the Line: Data Determines Loss-to-Loss Scaling Laws
>
> [3] - Improving Pretraining Data Using Perplexity Correlations
>
> **4. Computational Cost: Can you provide concrete wall-clock time measurements and memory overhead for 1B and 3B models? What percentage overhead compared to standard training?**
>
> We thank the reviewer for raising this point.  Architecture warm-up reduces memory usage during the early stages of training because only a subset of layers is active (160GB), and the peak memory converges to that of the standard setup (248GB) once all layers are unlocked. The wall-clock is the same as standard training time (18K TPS). We hope these measurements clarify that the method’s computational footprint remains close to that of standard training.

---

> ### Author Response · Authors · 2025-11-20
>
> **5. Gradient Clipping: How does your method compare against well-tuned gradient clipping? Can they be combined?**
>
> We thank the reviewer for raising this question. All of our experiments already use standard, well-tuned gradient clipping (global norm clipping at 1.0), so the reported improvements are on top of a clipped baseline rather than in place of it. Consistent with prior observations in large-scale LLM training, we find that gradient clipping improves robustness but does not by itself eliminate instabilities or edge-of-stability behaviour at the learning rates needed for efficient training.
>
> Conceptually, our method and gradient clipping act on different aspects of the dynamics: clipping controls gradient magnitude, whereas architecture warm-up modulates the effective curvature by controlling depth over time. As a result, they are complementary rather than mutually exclusive. In practice, they can be (and in our experiments already are) combined, with architecture warm-up providing additional stability and performance gains beyond what well-tuned gradient clipping alone achieves.
>
> **6. Initial Depth Selection: How sensitive are results to starting depth (1/3 vs 1/2 vs 2/3)? Can you provide a principled selection rule?**
>
> We thank the reviewer for raising this question.
>
> In our experiments we use half depth as the default starting point, and we think of this primarily as a stability–efficiency trade-off rather than a finely tuned hyperparameter. If the starting depth is too small (for example, around one third of the full depth), the resulting model during the high–learning-rate phase is very conservative: the effective network is substantially underparameterized, curvature remains safely within the stability margin, but the point at which the full capacity of the model is used is delayed. In practice, we find that this leads to slower optimization without clear stability benefits over the half-depth setting.
> Conversely, if the starting depth is too large (for example, around two thirds), the initial model is already close in curvature to the full-depth network. In that case, the training trajectory rapidly approaches the edge-of-stability regime at the peak learning rate, and the advantages of architecture warm-up are largely diminished: curvature spikes and instability patterns begin to resemble those of the baseline.
>
> Empirically, starting from half depth strikes a good balance: the model is shallow enough that the initial top eigenvalue of the Hessian stays comfortably within the stability margin at peak learning rate, but deep enough that we are not overly conservative and can exploit a substantial fraction of the model’s capacity early in training.
>
> **7. HVP Iterations: How does the required number of HVPs vary across training stages and model sizes? Should this be adjusted dynamically?**
>
> We did not observe that the HVP iterations need to be varied across training stages and model sizes. For all our experiments, we used 5 HVP iterations.
>
> **8. RMSNorm Initialization: What specifically goes wrong with RMSNorm from zero initialization? Did you try alternatives?**
>
> RMSNorm is known to work best when its scale parameter is initialized close to 1, which is also the default in the standard PyTorch implementation. In our experiments, we found that setting this scale to zero does not cause numerical instabilities or divergence, but it does significantly hurt optimization: the normalized activations are initially collapsed, leading to slower learning and worse final convergence (higher loss/perplexity) compared to the standard initialization. We therefore use the conventional “near-1” initialization in all our main experiments. Beyond comparing zero vs. the standard initialization, we did not find RMSNorm initialization to be a primary driver of the phenomena we study, and we view it as largely orthogonal to our main contribution.
>
> **9. Compute-Optimal: Can you provide compute-optimal comparisons with all baselines (not just QK-Norm)?**
>
> We have now added the compute optimal results for Softcap as well. We will add the results for the other baselines in the camera ready version.
>
> | Method       | Val PPL ↓ |
> |-------------|-----------|
> | QK-Norm     | 20.28     |
> | Softcap     | 32.44     |
> | Arch-Warmup | **18.35** |

---

> ### Author Response · Authors · 2025-11-20
>
> **10. LR Warm-up Replacement: Is Figure 6's result generalizable across model sizes and datasets, or specific to this setting?**
>
> For the LR Warm-up Replacement, the baseline is trained with a standard, well-tuned learning-rate schedule with warm-up. Concretely, we use commonly adopted hyperparameters for LLaMA-style models: peak learning rate $4 \times10^{−4}$, weight decay $0.1$, $2000$ warm-up steps, and a cosine decay schedule. These values are chosen following standard practice rather than being tailored to favour our method.
>
> The “architecture-warm-up–only” variant is obtained by taking this baseline configuration and removing only the LR warm-up: we set the number of warm-up steps to zero, keep the peak learning rate and decay schedule unchanged, keep all optimizer hyperparameters (including weight decay) fixed, and then enable architecture warm-up as described in the main text. The purpose of this experiment is to test whether a configuration that trains well under a standard LR schedule can still train stably once LR warm-up is removed, provided architecture warm-up is present.
>
> In the revised version, we have made this protocol explicit by (i) stating which hyperparameters are tuned for the baseline and over what ranges, (ii) clarifying which of these are held fixed when LR warm-up is removed, and (iii) emphasizing that the architecture-warm-up–only result should be interpreted as an existence proof of feasibility in one realistic configuration, not as a fully optimized recipe. Our main claims remain focused on the regime where architecture warm-up is combined with standard LR schedules, with the “replacement” experiment presented as an interesting direction rather than a universal statement.

---

### Official Review · Reviewer_y4WB · 2025-11-01

**Soundness:** 2
**Presentation:** 3
**Contribution:** 3
**Rating:** 6
**Confidence:** 3

**Summary:**

The paper makes two contributions. One is a to warm-up power iteration schemes to track the largest eigenvalue of the Hessian. The second is to propose architecture warm-up: unlocking more and more layers across training to account for natural increase in sharpness.
The warm-up scheme is quickly tested and we see that the largest eigenvalue of the Hessian evolves slow enough for the warmups to be efficient. The architecture warm-up appears to be an effective technique to improve stability during training. In particular the authors argue that it could replace the usual warm-up.

**Strengths:**

- The proposed tool is imple but could be of use. We hope that the authors will do their best to open source the code in all relevant platforms.
- The architecture warm-up is interesting. It is reminiscent of optimization techniques for deep learning that learned one layer at a time.
- Experiments are done at a relatively large scale.
- The improvements in stability are somewhat surprising. They also contribute to a very different approach to improve stability in training.

**Weaknesses:**

- The theoretical justifications are somewhat off: the spectrum is bounded above by something that depends on the number of layers. It is not bounded below. So these justifications are unclear.
- The experiments are strangely presented (see questions below). Many baselines diverge. One culprit seems to be the numver of warmup steps (2000) that does not look standard for the baselines.
- The authors claim that architecture warm-up could replace standard warm-up. Such a claim requires detailed experimental setup: how were all hyperparameters tuned (so learning rate, weight decay, warm-up steps).
- The authors claim that spacing of unlocking does not matter for the architecture warm-up but there must be extreme cases where it does. Unlocking every step would not work for example.
- The largest eigenvalue fo the Hessian generally drives dynamics mostly in the full batch case. The stochastic case may be governed by different mechanisms [1]. So even though architecture warm-up appears to work. It is unclear whether the authors have revealed why yet.
- Aggressive warm-up or even learning rate tuners can also directly enforce a smaller curvature [2]. So, if curvature is the leading reason for improved training curves, there could be other ways to drive the network towards more stable regions.

[1] Agarwala, A. and Pennington, J., 2024. High dimensional analysis reveals conservative sharpening and a stochastic edge of stability. arXiv preprint arXiv:2404.19261.
[2] Roulet, V., Agarwala, A., Grill, J.B., Swirszcz, G., Blondel, M. and Pedregosa, F., 2024. Stepping on the edge: Curvature aware learning rate tuners. Advances in Neural Information Processing Systems, 37, pp.47708-47740.

**Questions:**

- Why do the lambda max plots have a y-axis from 0 to 1? They could be larger than 1. And the plots should be in log-scale. If the plots were for example in log-scale or better centered, could we see to what value the preconditioned hessian converges (in the architecture warmup case)?
- Most importantly, can the authors provide a plot of the largest eigenvalue of the preconditioned hessian with architecture warm-up? We cannot see its evolution in the plots for now (too small). It would be great to see that evolution with the exact times when the architecture is unlocked and to which scale each time. It would be great to see whether progressive sharpening increases/decreases with the increased depth.
- The authors say line 366, "progressively unlock additional layers only when the effective curvature remains below within the stability margin". What does that mean? As long as the edge of stability is not reached, we could unlock the architecture. The sentence above does not reflect a trigger or a "only when".
- As said in the weakness section, it would be great to have the detailed experimental setup for figure 6: how were hyperparameters tuned, how robust is this observation, could there actual be limitations to architecture warm-up etc...

---

> ### Author Response · Authors · 2025-11-20
>
> We thank the reviewer for the insightful comments and suggestions. Please find our detailed response below.
>
> **1. The theoretical justifications are somewhat off: the spectrum is bounded above by something that depends on the number of layers. It is not bounded below. So these justifications are unclear.**
>
> The reviewer is correct that the upper bound in Eq. (14) is not sufficient to conclude that the curvature necessarily increases with the number of layers. This limitation is precisely what we highlight in the following Remark (L250), which we use to motivate the need for exact curvature tracking rather than relying on worst-case bounds. In particular, we explicitly state that *“Empirically, the curvature might be lower than the bound due to architecture components such as residual connections and normalization”* (L252), and that *“since the preconditioned geometry is the relevant one for adaptive optimizers, efficient online tracking of top curvature eigenvalues, especially the preconditioned analogue $\lambda_{max}(G_t)$, is more informative than pointwise analyses at initialization or at local optima”* (L254).
>
> In other words, the role of Eq. (14) and the accompanying remark is not to claim that curvature must grow with depth, but to show that such an upper bound can be derived from first principles and yet remains too weak to settle the question. This is exactly why our method focuses on online tracking of the curvature spectrum (and its preconditioned counterpart), which provides the practically relevant information that these bounds alone cannot guarantee.
>
> We apologize if the above point is not conveyed properly and have made it clearer in the revised version.
>
> **2. The experiments are strangely presented (see questions below). Many baselines diverge. One culprit seems to be the numver of warmup steps (2000) that does not look standard for the baselines**
>
> We appreciate the reviewer’s concern. However, the choice of 2000 warmup steps is not arbitrary: we follow the standard linear warmup schedule used in the original LLaMA pretraining setup (note that we also use a LLaMA-style decoder-only architecture) [1]. The same 2000-step warmup configuration is additionally reported in the AWS Trainium end-to-end LLaMA-2 pretraining recipe [2]. Since our experiments are based on the LLaMA architecture, we deliberately adopted this established schedule to keep the baselines faithful to standard practice.
>
> Note that a bulk of our experiments run for 5000 optimization steps (except  for compute optimal experiments in Appendix D, where we trained the models longer for 25B tokens). This is intentional: the goal of these experiments is to probe the high–learning-rate regime, where the stability threshold $2 / \eta$ is smallest and the models are most vulnerable to sharpness spikes and divergence. After roughly 5000 steps, the learning rate has already decayed substantially, the stability threshold increases. Extending every ablation far into this low–learning-rate phase would add significant compute cost while providing relatively little additional information about the stability properties we are interested in. In all cases, the same schedule (including the 2000-step warmup and 5000-step horizon) is applied identically to both the baselines and the architecture–warm-up variants.
>
> [1] - LLaMA: Open and Efficient Foundation Language Models
>
> [2] - https://aws.amazon.com/blogs/machine-learning/end-to-end-llm-training-on-instance-clusters-with-over-100-nodes-using-aws-trainium/

---

> ### Author Response · Authors · 2025-11-20
>
> **3. The authors claim that architecture warm-up could replace standard warm-up ... it would be great to have the detailed experimental setup for figure 6:**
>
> We thank the reviewer for raising this point.
>
> For the experiment in Sec. 4.6, the baseline is trained with a standard, well-tuned learning-rate schedule with warm-up. Concretely, we use commonly adopted hyperparameters for LLaMA-style models: peak learning rate $4 \times10^{−4}$, weight decay $0.1$, $2000$ warm-up steps, and a cosine decay schedule. These values are chosen following standard practice rather than being tailored to favour our method.
>
> The “architecture-warm-up–only” variant is obtained by taking this baseline configuration and removing only the LR warm-up: we set the number of warm-up steps to zero, keep the peak learning rate and decay schedule unchanged, keep all optimizer hyperparameters (including weight decay) fixed, and then enable architecture warm-up as described in the main text. The purpose of this experiment is to test whether a configuration that trains well under a standard LR schedule can still train stably once LR warm-up is removed, provided architecture warm-up is present.
>
> In the revised version, we have made this protocol explicit by (i) stating which hyperparameters are tuned for the baseline and over what ranges, (ii) clarifying which of these are held fixed when LR warm-up is removed, and (iii) emphasizing that the architecture-warm-up–only result should be interpreted as an existence proof of feasibility in one realistic configuration, not as a fully optimized recipe. Our main claims remain focused on the regime where architecture warm-up is combined with standard LR schedules, with the “replacement” experiment presented as an interesting direction rather than a universal statement.
>
> **4. The authors claim that spacing of unlocking does not matter for the architecture warm-up but there must be extreme cases where it does. Unlocking every step would not work for example.**
>
> The reviewer is correct that there could exist extreme schedules for unlocking layers (e.g., unlocking every step from the very beginning) that would be suboptimal or even fail in practice. Our claim is not that all conceivable schedules work, but rather that within a broad and practically relevant range, performance is not highly sensitive to the precise spacing.
>
> In particular, as long as the half-depth model is trained until the peak learning rate is reached, unlocking the remaining layers in a small number of stages, even all at once, does not harm convergence. This is because the newly enabled blocks are zero-initialized, so their contribution to the Jacobian/Hessian, and hence to the total (preconditioned) curvature, grows gradually from the well-conditioned half-depth baseline (Appendix E). The purpose of our empirical experiments is exactly to demonstrate this robustness: the method continues to converge well across a wide range of unlocking intervals, indicating that it does not rely on very granularly tuned warm-up schedules.

---

> ### Author Response · Authors · 2025-11-20
>
> **5. The largest eigenvalue fo the Hessian generally drives dynamics mostly in the full batch case. The stochastic case may be governed by different mechanisms.**
>
> This is an excellent point by the reviewer. We agree that, in general, stochastic gradient descent can exhibit different dynamics near the edge of stability compared to full-batch gradient descent. However, the literature also shows that this discrepancy shrinks rapidly as the batch size increases (typically once it exceeds ≈128), so that large-batch SGD is well-approximated by the full-batch theory.
> Concretely, the reviewer’s citation [1, Fig. 4] shows that for MNIST, once the batch size exceeds 128, the observed dynamics quickly converge to the full-batch predictions. Similarly, [2, Fig. 24] demonstrates that sharpness dynamics are accurately predicted by the full-batch theory for batch sizes larger than 128, for both MSE and cross-entropy losses.
>
> Typically, LLM training is performed with very large batch sizes (up to several thousand tokens per step. In our experiments we use a batch size of 1024. Moreover, for decoder-only Transformers with cross-entropy loss, the gradient at each step is an average over all tokens in all sequences in the batch. Thus, a batch of 1024 sequences of length 1024 corresponds to gradients averaged over ≈10⁶ token-level contributions, an effective batch size far beyond the $\approx 128$ regime where SGD dynamics are already well approximated by GD in prior work. For this reason, although SGD and GD can differ in principle, in our regime it is reasonable to use full-batch curvature intuitions to interpret and predict the observed behavior of architecture warm-up, which is what our analysis aims to do. In our experiments we also observe that
>
> $\log \bigl(\lambda_{\max}(G_t)\bigr) \approx 8.5$
>
> So that, at the peak learning rate $\eta = 0.008$
>
> $\lambda_{\max}(G_t)\eta \approx \exp(8.5) \cdot 0.008 \approx  39.3 \approx 38$
>
> which is the stability threshold for full batch Adam.
>
>
> [1] - High dimensional analysis reveals conservative sharpening and a stochastic edge of stability
>
> [2] -  Gradient Descent on Neural Networks Typically Occurs at the Edge of Stability
>
> **6. Aggressive warm-up or even learning rate tuners can also directly enforce a smaller curvature. So, if curvature is the leading reason for improved training curves, there could be other ways to drive the network towards more stable regions.**
>
> This is a good point, and we agree that aggressive learning-rate warm-up or LR tuning can, in effect, keep the effective curvature smaller and thereby improve stability. However, the question remains if that is an efficient way to achieve stability. For instance, in principle, one can always choose *\eta* small enough that the sharpness never approaches *2/\eta*, but such step sizes are suboptimal in terms of training speed and cost, sometimes dramatically so, especially in today’s extremely expensive LLM training regimes, where an iteration can cost thousands of dollars.
>
> In practice, recent large-scale model reports consistently show that, at reasonable step sizes and LR schedules, billion-parameter–scale models enter unstable regions and even diverge, a phenomenon we connect to the Edge-of-Stability regime using our mathematical tools. Thus, for standard LLMs trained at competitive learning rates, we believe the Edge-of-Stability regime should be viewed as the rule rather than the exception. Our goal is not to argue that curvature cannot be controlled by other means, but to show that (i) curvature is a key mechanism underlying the observed training behaviour, and (ii) architecture warm-up offers a complementary way to steer the optimization into more stable regions without resorting to conservative learning rates.

---

> ### Author Response · Authors · 2025-11-20
>
> **7.  If the plots were for example in log-scale or better centered, could we see to what value the preconditioned hessian converges (in the architecture warmup case)? Most importantly, can the authors provide a plot of the largest eigenvalue of the preconditioned hessian with architecture warm-up? We cannot see its evolution in the plots for now (too small). It would be great to see that evolution with the exact times when the architecture is unlocked and to which scale each time. It would be great to see whether progressive sharpening increases/decreases with the increased depth.**
>
> We thank the reviewer for this suggestion. Following the reviewer’s feedback, we have added an additional plot in the revised version that shows the unnormalized largest eigenvalue of the preconditioned Hessian on a logarithmic scale (Fig. 10). In this new plot, for the architecture–warm-up run we observe that
>
> $\log\bigl(\lambda_{\max}(G_t)\bigr) \approx 8.5$
>
> so that, at the peak learning rate $\eta = 0.008$
>
> $\lambda_{\max}(G_t) \eta \approx \exp(8.5) \cdot 0.008 \approx  39.3 \approx 38$
>
> which matches the stability threshold for adaptive methods reported in the literature. In contrast, the baseline run lies well above this threshold.
>
> It is clear that architecture warm-up keeps the preconditioned curvature close to the theoretical stability margin, whereas the baseline substantially exceeds it, thereby providing a more direct illustration of the stability gains achieved through architecture warm-up. However, we do not see progressive unlocking of layers causing any discontinuity or stark jumps in the Hessian evolution, perhaps due to the smooth effective depth growth achieved by zero initialization.
>
> **8. The authors say line 366, "progressively unlock additional layers only when the effective curvature remains below within the stability margin". What does that mean? As long as the edge of stability is not reached, we could unlock the architecture. The sentence above does not reflect a trigger or a "only when".**
>
> We thank the reviewer for pointing out this ambiguity. Our intention was not to claim that we use an explicit, curvature-based trigger to decide when to unlock layers. Rather, the sentence was meant to provide intuition: as the learning rate decays, the Edge-of-Stability (EoS) threshold $2/\eta$ increases, so the effective curvature is further within the stability margin, and this is the regime in which we schedule the unlocking of additional layers.
>
> In other words, we use a fixed schedule that places the unlocking events in the LR-decay region, where our curvature measurements indicate that the dynamics are safely away from the EoS boundary. We have revised the wording in the updated version to make this clearer and to remove the misleading “only when” phrasing that suggested a hard trigger.

---

> ### Comment · Reviewer_y4WB · 2025-11-24
> **Thanks for the answers**
>
> Thank you for all your answers. A few additional questions:
>
> - Can you detail the setup for plotting Fig. 10? Typically the learning rate of the baseline.
>
> - Why are the baselines diverging in the first place? (as in Fig. 10) It's a setup with a too high learning rate? The authors could consider a plot "final loss" vs "learning rate" with or without architecture warmup (a bit like the sensibility analyses done in [1]). This would show clearly how larger is the range of valid learning rates using architecture warmup and it would also show that it can enable better final losses.
>
> [1] Small-scale proxies for large-scale Transformer training instabilities, Wortsman et al, ICLR 2024

---

> ### Author Response · Authors · 2025-11-26
>
> We thank the reviewer for the follow up questions.
>
> **Can you detail the setup for plotting Fig. 10? Typically the learning rate of the baseline.**
>
> We use a peak learning rate of $8 \times 10^{-3}$, context length 1024, embedding dimension 2048, 32 heads, global batch size 1024, weight decay 0.01, AdamW optimizer with standard parameters, and a linear warmup over 2000 steps followed
> by linear decay.
>
> **Why are the baselines diverging in the first place? (as in Fig. 10) It's a setup with a too high learning rate? The authors could consider a plot "final loss" vs "learning rate" with or without architecture warmup (a bit like the sensibility analyses done in [1]). This would show clearly how larger is the range of valid learning rates using architecture warmup and it would also show that it can enable better final losses.**
>
> We thank the reviewer for the suggestion.  The baseline diverges because, at this peak learning rate, the effective curvature exceeds the stability boundary: the product $\lambda_{max}(H)\eta$ crosses the edge-of-stability threshold, leading to sharp loss spikes and eventual divergence. As we previously noted one can always reduce $\eta$ to stay within a stable regime. However, doing so comes at a substantial cost in convergence speed and, at LLM scale, in monetary cost. Moreover, as reported in several large-model training reports (e.g., LLaMA, OLMo, PaLM, Gemini, Kimi K2 etc.), even “reasonable’’ learning rates often produce loss spikes and instabilities in deep models, consistent with the rapid growth of curvature with depth that we report.
>
> Inline with the reviewer’s suggestion, we have added a plot (Please see *torchtitan-archwarmup-arch-warmup-clean\plots\Lr_stability.png* in supplementary zip file) that shows that increasing the learning rate initially improves convergence, but beyond a certain threshold the baseline becomes unstable and diverges, while architecture warm-up continues to train stably at those learning rates (Fig. 4 in the paper). In other words, architecture warm-up does not merely “allow a higher LR’’ in a vacuous sense; it effectively *enlarges the stable and performant learning-rate region*, which is precisely the regime of practical interest at large scale.

---

### Meta-Review · Area_Chair_4jrG · 2026-01-07

**Summary:**

Overall, the reviewers agree that the paper introduces an **architecture warm-up strategy** that addresses a real and important challenge in large-scale Transformer training, namely training instability caused by curvature and edge-of-stability effects. There is **broad agreement among the reviewers** on the relevance and significance of this contribution. Reviewer y4WB, Reviewer 4tpE, and Reviewer KvM5 each assigned the paper a score of **6**, and both Reviewer y4WB and Reviewer KvM5 explicitly indicated that they were generally in favor of acceptance in their responses to Area Chair UJST during the **Discussion Phase on 27 Nov 2025**.

**Reviewer Concerns:**

**Reviewer y4WB’s concerns:**

The main concerns regarding the theoretical justification of the spectral bounds, the detailed experimental setup (including diverged baselines), ambiguities in the claims, as well as several additional questions raised during the rebuttal, were addressed.

**Reviewer 4tpE’s concerns:**

She/he concerns were mainly related to the experiments. Following the reviewer’s suggestions, the authors have added the corresponding experimental results.

**Reviewer KvM5’s concerns:**

She/he engaged with the authors during the rebuttal and raised concerns regarding whether architecture warmup offers benefits beyond LR warmup (especially when depth and width scale together), the relationship between curvature control and higher learning rates, and the minibatch curvature measurements. After the authors clarified these points and added batchsize robustness experiments, Reviewer KvM5 explicitly thanked authors and stated that additional experiments addressed concerns.

The remaining outstanding concern is a broader question regarding the impact on downstream tasks and fine-tuning, which was acknowledged as an interesting direction for future work.

**Reviewer Scores:**

Although **Reviewer y4WB** and **Reviewer KvM5** explicitly indicated that they were generally in favor of acceptance in response to Area Chair UJST during the **Discussion Phase on 27 Nov 2025**, they did not indicate any intention to increase their scores. Therefore, I believe that Reviewer y4WB and Reviewer KvM5 would maintain their original scores.

**Reviewer 4tpE**’s concerns were mainly related to the experiments. Following the reviewer’s suggestions, the authors have added the corresponding experimental results. As a result, I think Reviewer 4tpE may increase the score (likely from 6 to 8) if She/he had been able to participate fully in the discussion.

---

### Decision · Program_Chairs · 2026-01-26

Accept (Poster)